# LDC-MTL: Balancing Multi-Task Learning through Scalable Loss Discrepancy Control

## Abstract

Multi-task learning (MTL) has been widely adopted for its ability to simultaneously learn multiple tasks. While existing gradient manipulation methods often yield more balanced solutions than simple scalarization-based approaches, they typically incur a significant computational overhead of $\mathcal{O}(K)$ in both time and memory, where $K$ is the number of tasks. In this paper, we propose LDC-MTL, a simple and scalable loss discrepancy control approach for MTL, formulated from a bilevel optimization perspective. Our method incorporates two key components: (i) a bilevel formulation for fine-grained loss discrepancy control, and (ii) a scalable first-order bilevel algorithm that requires only $\mathcal{O}(1)$ time and memory. Theoretically, we prove that LDC-MTL guarantees convergence not only to a stationary point of the bilevel problem with loss discrepancy control but also to an $\epsilon$-accurate Pareto stationary point for all $K$ loss functions under mild conditions. Extensive experiments on diverse multi-task datasets demonstrate the superior performance of LDC-MTL in both accuracy and efficiency.

## 1 Introduction

In recent years, Multi-Task Learning (MTL) has received increasing attention for its ability to predict multiple tasks simultaneously using a single model, thereby reducing computational overhead. This versatility has enabled a wide range of applications, including autonomous driving (Chen et al., 2018), recommendation systems (Wang et al., 2020), and natural language processing (Zhang et al., 2022).

One of the main challenges in MTL is the imbalance and discrepancy in task losses, where different tasks progress at uneven rates during training. This discrepancy stems from several sources: variations in loss magnitudes due to differing units or scales (e.g., meters vs. millimeters) (Kendall et al., 2018; Liu et al., 2019), heterogeneity in task types (e.g., regression vs. classification) (Dai et al., 2023; Lin et al., 2021), and conflicting gradient directions across tasks (Yu et al., 2020; Liu et al., 2021a). When unaddressed, such discrepancies may cause certain tasks to dominate the optimization trajectory, ultimately leading to degraded performance on others.

To mitigate this issue, MTL research generally follows two main paradigms. The first is the class of scalarization-based methods, which transform MTL into a single-objective optimization problem by aggregating task losses, typically through weighted or averaged sums. Early works adopted static weighting schemes for their simplicity and scalability (Caruana, 1997), but these often led to degraded multi-task performance relative to single-task baselines, largely due to persistent gradient conflicts under fixed weights (Xiao et al., 2024). As a remedy, more recent approaches explore dynamic loss weighting strategies that adapt during training (Kendall et al., 2018; Liu et al., 2019; Lin et al., 2021; Dai et al., 2023). However, these methods do not explicitly address loss discrepancy, allowing task interference to persist and leading to imbalanced performance across tasks. The second line of work involves gradient manipulation techniques, which aim to promote balanced optimization by explicitly resolving gradient conflicts. These approaches seek update directions that are more equitable across tasks (Désidéri, 2012; Liu et al., 2021a; Ban & Ji, 2024; Navon et al., 2022; Yu et al., 2020; Fernando et al., 2023; Xiao et al., 2024). While often effective in reducing task interference, they typically require computing and storing gradients from all $K$ tasks at each iteration, incurring $O(K)$ time and memory costs. This scalability bottleneck poses challenges for large-scale MTL scenarios involving deep architectures and massive datasets.

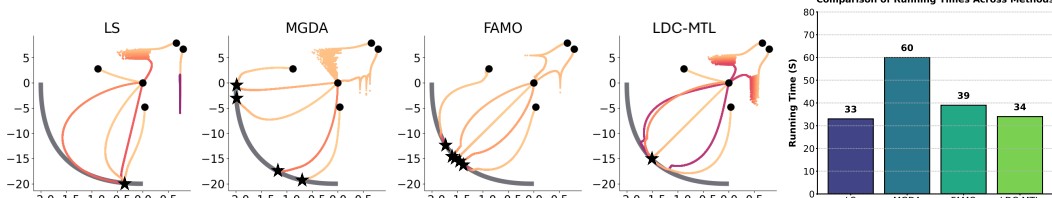

Figure 1: The loss trajectories of a toy 2-task learning problem from Liu et al. (2024) and the runtime comparison of different MTL methods for 50000 steps. Stars on the Pareto front denote the convergence points. Although FAMO (Liu et al., 2024) achieves more balanced results than Linear Scalarization (LS) and MGDA (Désidéri, 2012), it converges to different points on the Pareto front. LDC-MTL reaches the same balanced point with a computational cost comparable to the LS. Full experimental details can be found in Appendix A.3.

In this paper, we propose a simple and scalable loss discrepancy control approach for MTL from a novel bilevel optimization perspective. Our approach comprises two key components: a bilevel formulation for fine-grained loss discrepancy control, and a scalable first-order bilevel algorithmic design. Our specific contributions are summarized as follows.

- **Bilevel formulation for loss discrepancy control.** At the core of our bilevel formulation, the lower-level problem optimizes the model parameters by minimizing a weighted sum of individual loss functions. Meanwhile, the upper-level problem adjusts these weights to minimize the discrepancies among the loss functions, ensuring balanced learning across tasks.

- **Efficient algorithms with $O(1)$ time and memory cost.** We develop Loss Discrepancy Control for Multi-Task Learning (LDC-MTL), a highly efficient algorithm tailored to solve the proposed bilevel problem with loss discrepancy control. Unlike traditional bilevel methods, LDC-MTL has a fully single-loop structure without second-order gradient computations, resulting in an overall $\mathcal{O}(1)$ time and memory complexity. The 2-task toy example in Figure 1 illustrates that our LDC-MTL method achieves a more balanced solution compared to other competitive approaches while maintaining superior computational efficiency.

- **Empirical performance.** Extensive experiments show that our proposed LDC-MTL method outperforms a wide range of scalarization-based and gradient manipulation approaches across multiple supervised multi-task datasets, including QM9 (Ramakrishnan et al., 2014), CelebA (Liu et al., 2015), and Cityscapes (Cordts et al., 2016), while also demonstrating superior efficiency and scalability.

- **Experimental analysis.** We conduct a deeper exploration showing that task losses under LDC-MTL become more concentrated and consistently lower. As a side effect of controlling loss discrepancies, gradient conflicts are also reduced. In addition, when compared to weight-swept linear scalarization (LS), LDC-MTL solutions align with the Pareto frontier (PF) and achieve consistently better results. Our experiments also suggest that the dynamic training trajectory, rather than just the final model parameters, plays a key role in the strong performance of our method. Overall, these results highlight the importance of dynamic loss discrepancy control in LDC-MTL.

- **Theoretical guarantees.** Theoretically, we show that LDC-MTL guarantees convergence not only to a stationary point of the bilevel problem with loss discrepancy control but also to an $\epsilon$-accurate Pareto stationary point for all $K$ individual loss functions under suitable conditions.

## 2    RELATED WORKS

**Multi-task learning.** MTL has recently garnered significant attention in practical applications. One line of research focuses on model architecture, specifically designing various sharing mechanisms (Kokkinos, 2017; Ruder et al., 2019). Another direction addresses the mismatch in loss magnitudes across tasks, proposing methods to balance them. For example, Kendall et al. (2018) balanced tasks by weighting loss functions based on homoscedastic uncertainties, while Liu et al. (2019) dynamically adjusted weights by considering the rate of change in loss values for each task.

Besides, one prominent approach frames MTL as a Multi-Objective Optimization (MOO) problem. Sener & Koltun (2018) introduced this perspective in deep learning, inspiring methods based on the Multi-Gradient Descent Algorithm (MGDA) (Désidéri, 2012). Subsequent work has aimed to address gradient conflicts. For instance, PCGrad (Yu et al., 2020) resolves conflicts by projecting gradients onto the normal plane, GradDrop (Chen et al., 2020) randomly drops conflicting gradients, and CAGrad (Liu et al., 2021a) constrains update directions to balance gradients. Additionally, Nash-MTL (Navon et al., 2022) formulates MTL as a bargaining game among tasks, while FairGrad (Ban & Ji, 2024) incorporates $\alpha$-fairness into gradient adjustments. Achituve et al. (2024) introduces a novel gradient aggregation approach using Bayesian inference to reduce the running time.

Prior works (Kurin et al., 2022; Xin et al., 2022) have shown that several gradient-based MTL methods fail to outperform linear scalarization (LS) with weight sweeping. In contrast, our extensive experiments demonstrate that our method yields solutions that dominate those obtained by other methods in Figure 5. On the theoretical side, Zhou et al. (2022) analyzed the convergence properties of stochastic MGDA, and Fernando et al. (2023) proposed a method to reduce bias in the stochastic MGDA with theoretical guarantees. More recent advancements include a double-sampling strategy with provable guarantees introduced by Xiao et al. (2024) and Chen et al. (2024).

**Bilevel optimization.** Bilevel optimization, first introduced by Bracken & McGill (1973), has been extensively studied over the past few decades. Early research primarily treated it as a constrained optimization problem (Hansen et al., 1992; Shi et al., 2005). More recently, gradient-based methods have gained prominence due to their effectiveness in machine learning applications. Many of these approaches approximate the hypergradient using either linear systems (Domke, 2012; Ji et al., 2021) or automatic differentiation techniques (Maclaurin et al., 2015; Franceschi et al., 2017). However, these methods become impractical in large-scale settings due to their significant computational cost (Xiao & Ji, 2023; Yang et al., 2024b). The primary challenge lies in the high cost of gradient computation: approximating the Hessian-inverse vector requires multiple first- and second-order gradient evaluations, and the nested sub-loops exacerbate this inefficiency. To address these limitations, recent studies have focused on reducing the computational burden of second-order gradients. For example, some methods reformulate the lower-level problem using value-function-based constraints and solve the corresponding Lagrangian formulation (Kwon et al., 2023; Yang et al., 2024a). The work studies convex bilevel problems and proposes a zeroth-order optimization method with finite-time convergence to the Goldstein stationary point (Chen et al., 2023). Meanwhile, several works investigate multi-objective bilevel optimization problems, but their computational cost remains high becausee of the inner iterations or dependence on Hessian terms (Ye et al., 2021; 2024). In this work, we propose a simplified first-order bilevel method for MTL, motivated by intriguing empirical findings.

## 3 PRELIMINARY

**Scalarization-based methods.** MTL aims to optimize multiple tasks (objectives) simultaneously with a single model. The straightforward approach is to optimize a weighted summation of all loss functions: $\min_x L_{total}(x) = \sum_{i=1}^K w_i l_i(x)$, where $x \in \mathbb{R}^d$ denotes the model parameter, $l_i(x) : \mathbb{R}^d \to \mathbb{R}_{\geq 0}$ represents the loss function of the $i$-th task and $K$ is the number of tasks. This approach faces three key challenges: 1) loss values could differ in scale, 2) fixed weights can lead to significant gradient conflicts, potentially allowing one task to dominate the learning process (Xiao et al., 2024; Wang et al., 2024); and 3) the overall performance is highly sensitive to the weighting of different losses (Kendall et al., 2018). Consequently, such methods often struggle with performance imbalances across tasks.

**Gradient manipulation methods.** To mitigate gradient conflicts, gradient manipulation methods dynamically compute an update $d^t$ at each epoch to balance progress across tasks, where $t$ is the epoch index. The update $d^t$ is typically a convex combination of task gradients, expressed as:

$$d^t = G(x^t)w^t, \ \ \text{where } w^t = h(G(x^t)),$$

with $G(x^t) = [\nabla l_1(x^t), \nabla l_2(x^t), \ldots, \nabla l_K(x^t)]^\top$. The weight vector $w^t$ is determined by a function $h(\cdot) : \mathbb{R}^{K \times d} \to \mathbb{R}^K$, which varies depending on the specific method. However, these methods often require computing and storing the gradients of all $K$ tasks during each epoch, making them less scalable and resource-intensive, particularly in large-scale scenarios. Therefore, it is necessary to develop lightweight methods that achieve balanced performance.

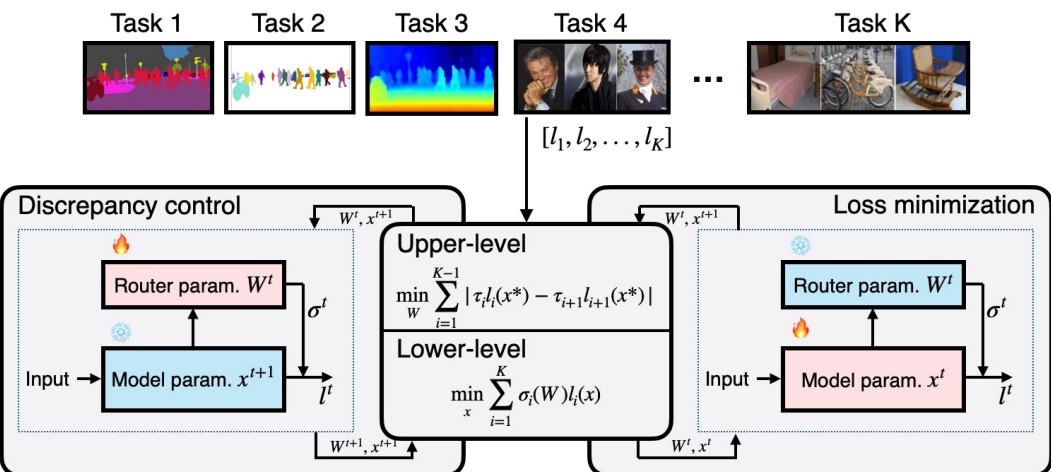

Figure 2: Our bilevel loss discrepancy control pipeline for multi-task learning. First, different task losses will be computed. Then, the lower-level problem optimizes the model parameter $x^t$ by minimizing the weighted sum of task losses, and the upper-level problem optimizes the router model parameter $W^t$ for fine-grained loss discrepancy control.

**Pareto concepts.** Solving the MTL problem is challenging because it is difficult to identify a common $x$ that achieves the optima for all tasks. Instead, a widely accepted target is finding a Pareto stationary point. Suppose we have two points $x_1$ and $x_2$. It is claimed that $x_1$ dominates $x_2$ if $l_i(x_1) \leq l_i(x_2)\ \forall i \in [K]$, and $\exists j\ l_j(x_1) < l_j(x_2)$. A point is Pareto optimal if it is not dominated by any other point, implying that no task can be improved further without sacrificing another. Besides, a point $x$ is a Pareto stationary point if $\min_{w \in \mathcal{W}} \|G(x)w\| = 0$.

## 4 LOSS DISCREPANCY CONTROL FOR MULTI-TASK LEARNING

In this section, we present our bilevel loss discrepancy control framework for multi-task learning. As illustrated in Figure 2, this framework contains a fine-grained bilevel loss discrepancy control procedure and a simplified first-order optimization pipeline. We first introduce the motivation and high-level idea that guide the overall design before moving to the technical details.

### 4.1 MOTIVATION AND HIGH-LEVEL IDEA

Loss discrepancy commonly arises in MTL for several reasons. First, loss functions may operate on different scales due to task heterogeneity (e.g., classification vs. regression, as in QM9 (Ramakrishnan et al., 2014)) or because they are measured in varying units (e.g., meters, centimeters, or millimeters; (Kendall et al., 2018)). Second, losses can evolve at different rates depending on task difficulty (Si et al., 2025). As a result, although all task losses are expected to converge toward zero, their convergence speeds are often imbalanced when treated with equal importance.

To mitigate this issue, some prior approaches, e.g., (Si et al., 2025), introduce a ranking-based strategy that optimizes only a subset of tasks with the largest current losses. While effective in reducing discrepancy, such methods require solving an additional non-smooth optimization problem, which limits scalability. Moreover, Roh et al. (2020) proposes to reduce loss gaps across different groups to enhance model fairness. In preference-based MOO, the goal is to identify a Pareto-optimal point such that $p_1 l_1 = p_2 l_2 = \cdots = p_K l_K$, where $p_1, \ldots, p_K$ denote user-specified preferences. In balanced MTL, these preferences are naturally equal, reducing the goal to aligning task losses so that they remain close to one another.

Motivated by this perspective, we explore a natural question: *can loss discrepancy in MTL be addressed by explicitly reducing the mutual gaps among task losses, while simultaneously ensuring that the solutions remain sufficiently close to the Pareto-stationary set?* In the next section, we provide a positive solution through a bilevel optimization formulation.

## 4.2 BILEVEL FORMULATION FOR LOSS DISCREPANCY CONTROL

Building on this motivation, we propose a bilevel optimization-based approach, where the upper-level problem aims to mitigate the loss discrepancy, while the lower-level problem ensures that the solution remains within the Pareto-stationary set. Specifically, we consider the following formulation:

$$\min_{W} \sum_{i=1}^{K-1} |\tau_i l_i(x^*) - \tau_{i+1} l_{i+1}(x^*)| := f(W, x^*)$$

$$\text{s.t. } x^* \in \arg\min_{x} \sum_{i=1}^{K} \sigma_i(W) l_i(x) := g(W, x), \tag{1}$$

where we denote $x^* = x^*(W)$ for notational convenience. Note that we define a routing function $\sigma(W) \in \mathbb{R}^K$, which is parameterized by a small neural network with a softmax output layer. It takes the shared feature as the input and the output weights $K$ for different tasks. For the upper-level weight vector $\tau = (\tau_1, ..., \tau_K)$, it controls the loss discrepancy, and we provide two effective options: (i) $\tau = \sigma(W)$ and (ii) $\tau = \mathbf{1}$ that work well in experiments. It can be seen from eq. (1) that the lower-level problem minimizes the weighted sum of losses w.r.t. the model parameters $x$, while the upper-level problem minimizes the accumulated weighted loss gaps w.r.t. the parameters $W$, controlling the loss discrepancy among tasks. Notably, the optimal solution does not require all losses to be equal, and there will be a **trade-off** between loss minimization and loss discrepancy control, as shown in Remark 1.

## 4.3 SCALABLE FIRST-ORDER ALGORITHM DESIGN

To enable large-scale applications, we adopt an efficient first-order method to solve the problem in eq. (1). Inspired by recent advances in first-order bilevel optimization (Kwon et al., 2023; Yang et al., 2023), we reformulate it into an equivalent constrained optimization problem as follows.

$$\min_{W,x} f(W, x) \text{ s.t. } \underbrace{\sum_{i=1}^{K} \sigma_i(W) l_i(x) - \sum_{i=1}^{K} \sigma_i(W) l_i(x^*)}_{\text{penalty function } p(W,x)} \leq 0.$$

Then, given a penalty constant $\lambda > 0$, penalizing $p(W, x)$ into the upper-level loss function yields

$$\min_{W,x} f(W, x) + \lambda \sum_{i=1}^{K} \Big( \sigma_i(W) l_i(x) - \sigma_i(W) l_i(x^*) \Big). \tag{2}$$

Intuitively, a larger $\lambda$ allows more precise training on model parameters $x$ such that $x$ converges closer to $x^*$. Conversely, a smaller $\lambda$ prioritizes upper-level loss discrepancy control during training. The main challenge of solving the penalized problem above lies in the updates of $W$, as shown below:

$$W^{t+1} = W^t - \alpha\Big(\nabla_W f(W^t, x^t) + \lambda\big(\nabla_W g(W^t, x^t) - \nabla_W g(W^t, z_N^t)\big)\Big), \tag{3}$$

where $t$ is the epoch index, $\alpha$ is the step size, and $z_N^t$ is an approximation of $x_t^* \in \arg\min_x g(W^t, x)$ through the following loop of $N$ iterations each epoch.

$$z_{n+1}^t = z_n^t - \beta\nabla_z g(W^t, z_n^t), n = 0, 1, ...N-1, \tag{4}$$

where $N$ is typically chosen to be sufficiently large, ensuring that $z_N^t$ closely approximates $x_t^*$ (the full algorithm is provided in Algorithm 2 in the appendix). Consequently, this sub-loop of iterations incurs significant computational overhead, driven by the high dimensionality of $z$ (matching that of the model parameters) and the large value of $N$.

---

**Algorithm 1: LDC-MTL**

**Initialize:** $W^0, x^0$
**for** $t = 0, 1, \ldots, T-1$ **do**
  $x^{t+1} = x^t - \alpha\left(\nabla_x f(W^t, x^t) + \lambda\nabla_x g(W^t, x^t)\right)$
  $W^{t+1} = W^t - \alpha\left(\nabla_W f(W^t, x^t) + \lambda\nabla_W g(W^t, x^t)\right)$
**end for**

---

**Remark 1** *This formulation encourages task losses to be close but not necessarily equal. The penalty constant $\lambda$ controls the trade-off between minimizing the sum of weighted losses and reducing the loss discrepancy among tasks.*

**Scalable Algorithm.** Our experiments show that the gradient norm $\|\nabla_W g(W^t, z_N^t)\|$ remains small, typically orders of magnitude smaller than the gradient norm $\|\nabla_W g(W^t, x^t)\|$, which is used to update the outer parameters $W$. This behavior is illustrated in Figure 3 and Figure 7 in the Appendix. Specifically, we set $N = 50$ during training. On average, the ratio $\|\nabla_W g(W^t, x^t)\|/\|\nabla_W g(W^t, z_N^t)\|$ exceeds 100, despite some fluctuations. Under these conditions, the term $\nabla_W g(W, z_N^t)$ can be safely neglected, thereby eliminating the need for the expensive loop in eq. (4). This approximation has been effectively utilized in large-scale applications, such as fine-tuning large language models, to reduce memory and computational costs (Shen

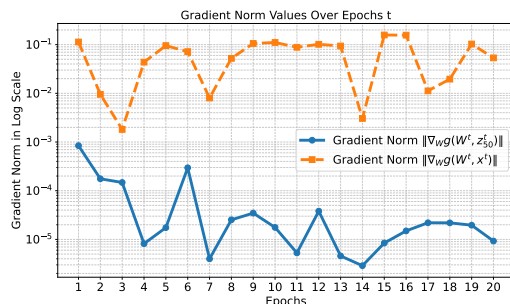

Figure 3: Gradient norm values during the training process on the Cityscapes dataset. Similar phenomena have also been observed in other datasets.

et al., 2024a). It also serves as a foundation for our proposed algorithm, Loss Discrepancy Control for Multi-Task Learning (LDC-MTL), described in Algorithm 1. LDC-MTL employs a fully single-loop structure, which requires only a single gradient computation for both variables per epoch, resulting in a $\mathcal{O}(1)$ time and memory cost. In Section 6, we show that our LDC-MTL method attains both an $\epsilon$-accurate stationary point for the bilevel problem in eq. (1) and an $\epsilon$-accurate Pareto stationary point for the original loss functions under mild conditions.

## 5 EMPIRICAL RESULTS

In this section, we conduct extensive practical experiments under multitask classification, regression, and mixed settings to demonstrate the effectiveness of our method. More experimental details can be found in Appendix A. All experiments are conducted on one NVIDIA A6000.

**Baselines and evaluation.** To demonstrate the effectiveness of our proposed method, we evaluate its performance against a broad range of baseline approaches. The compared methods include scalarization-based algorithms, such as Linear Scalarization (LS), Scale-Invariant (SI), Random Loss Weighting (RLW) (Lin et al., 2021), Dynamic Weight Average (DWA) (Liu et al., 2019), Uncertainty Weighting (UW) (Kendall et al., 2018), FAMO (Liu et al., 2024), and GO4Align (Shen et al.,

Table 1: Results on CelebA (40-task), QM9 (11-task), and NYU-v2 (3-task) datasets. The best results are highlighted in **bold**, while the second-best results are indicated with underlines.

| Method | CelebA | | QM9 | NYU-v2 | Cost |
|---|---|---|---|---|---|
| | $\Delta m\% \downarrow$ | MR | $\Delta m\% \downarrow$ | $\Delta m\% \downarrow$ | |
| MGDA (Désidéri, 2012) | 14.85 | 12.03 | 120.5 | 1.38 | |
| PCGrad (Yu et al., 2020) | 3.17 | 7.92 | 125.7 | 3.97 | |
| CAGrad (Liu et al., 2021a) | 2.48 | 7.35 | 112.8 | 0.20 | |
| IMTL-G (Liu et al., 2021b) | 0.84 | 5.75 | 77.2 | -0.76 | $\mathcal{O}(K)$ |
| Nash-MTL (Navon et al., 2022) | 2.84 | 6.17 | 62.0 | -4.04 | |
| FairGrad (Ban & Ji, 2024) | 0.37 | 5.67 | 57.9 | -4.66 | |
| ConsMTL (Qin et al., 2025) | **-1.42** | 2.75 | 23.2 | **-6.72** | |
| LS | 4.15 | 7.47 | 177.6 | 5.59 | |
| SI | 7.20 | 8.97 | 77.8 | 4.39 | |
| RLW (Lin et al., 2021) | 1.46 | 6.15 | 203.8 | 7.78 | |
| DWA (Liu et al., 2019) | 3.20 | 8.12 | 175.3 | 3.57 | $\mathcal{O}(1)$ |
| UW (Kendall et al., 2018) | 3.23 | 6.90 | 108.0 | 4.05 | |
| FAMO (Liu et al., 2024) | 1.21 | 5.85 | 58.5 | -4.10 | |
| GO4Align (Shen et al., 2024b) | 0.88 | N/A | 52.7 | -6.08 | |
| **LDC-MTL (ours)** | -1.31 | 2.62 | 49.5 | -4.40 | |

2024b). We also benchmark against gradient manipulation methods, including Multi-Gradient Descent Algorithm (MGDA) (Désidéri, 2012), PCGrad (Yu et al., 2020), GradDrop (Chen et al., 2020), CAGrad (Liu et al., 2021a), IMTL-G (Liu et al., 2021b), MoCo (Fernando et al., 2023), Nash-MTL (Navon et al., 2022), FairGrad (Ban & Ji, 2024), and ConsMTL (Qin et al., 2025). To provide a comprehensive evaluation, we report the performance of each individual task and employ one additional metric: $\mathbf{\Delta m}\%$ to quantify overall performance (Maninis et al., 2019). The $\mathbf{\Delta m}\%$ metric measures the average relative performance drop of a multi-task model compared to its corresponding single-task learning (STL). Formally, it is defined as: $\Delta m\% = \frac{1}{K} \sum_{i=1}^{K} (-1)^{\delta_k} (M_{m,k} - M_{b,k})/M_{b,k} \times 100$, where $M_{m,k}$ and $M_{b,k}$ represent the performance of the $k$-th task for the multi-task model $m$ and single-task model $b$, respectively. The indicator $\delta_k = 1$ if higher values indicate better performance and 0 otherwise.

Table 2: Results on Cityscapes (2-task) dataset. Each experiment is repeated 3 times with different random seeds, and the average is reported. The best results are highlighted in **bold**, while the second-best results are indicated with underlines. *Following prior works Liu et al. (2021a); Fernando et al. (2023); Xiao et al. (2024); Ban & Ji (2024), we report the mean values of $\Delta m\%$ for all results in the main text, with standard deviations provided in Appendix A.*

| METHOD | SEGMENTATION | | DEPTH | | $\Delta m\% \downarrow$ | MR | COST |
|---|---|---|---|---|---|---|---|
| | MIOU ↑ | PIX ACC ↑ | ABS ERR ↓ | REL ERR ↓ | | | |
| STL | 74.01 | 93.16 | 0.0125 | 27.77 | | | |
| MGDA (DÉSIDÉRI, 2012) | 68.84 | 91.54 | 0.0309 | 33.50 | 44.14 | 13.0 | |
| PCGRAD (YU ET AL., 2020) | 75.13 | 93.48 | 0.0154 | 42.07 | 18.29 | 10.75 | |
| GRADDROP (CHEN ET AL., 2020) | 75.27 | 93.53 | 0.0157 | 47.54 | 23.73 | 8.75 | |
| CAGRAD (LIU ET AL., 2021A) | 75.16 | 93.48 | 0.0141 | 37.60 | 11.64 | 8.0 | |
| IMTL-G (LIU ET AL., 2021B) | 75.33 | 93.49 | 0.0135 | 38.41 | 11.10 | 6.5 | $\mathcal{O}(K)$ |
| MOCO (FERNANDO ET AL., 2023) | 75.42 | 93.55 | 0.0149 | 34.19 | 9.90 | 5.5 | |
| NASH-MTL (NAVON ET AL., 2022) | 75.41 | 93.66 | 0.0129 | 35.02 | 6.82 | 4.0 | |
| FAIRGRAD (BAN & JI, 2024) | **75.72** | **93.68** | 0.0134 | 32.25 | 5.18 | 3.0 | |
| CONSMTL (QIN ET AL., 2025) | 75.57 | 93.32 | 0.0131 | **26.41** | **-0.59** | 2.25 | |
| LS | 75.18 | 93.49 | 0.0155 | 46.77 | 22.60 | 11.0 | |
| SI | 70.95 | 91.73 | 0.0161 | 33.83 | 14.11 | 16.25 | |
| RLW (LIN ET AL., 2021) | 74.57 | 93.41 | 0.0158 | 47.79 | 24.38 | 13.5 | |
| DWA (LIU ET AL., 2019) | 75.24 | 93.52 | 0.0160 | 44.37 | 21.45 | 13.25 | $\mathcal{O}(1)$ |
| UW (KENDALL ET AL., 2018) | 72.02 | 92.85 | 0.0140 | 30.13 | 5.89 | 10.0 | |
| FAMO (LIU ET AL., 2024) | 74.54 | 93.29 | 0.0145 | 32.59 | 8.13 | 8.75 | |
| GO4ALIGN (SHEN ET AL., 2024B) | 72.63 | 93.03 | 0.0164 | 27.58 | 8.11 | 11.75 | |
| **LDC-MTL (OURS)** | 74.53 | 93.42 | **0.0128** | 26.79 | -0.57 | 5.5 | |

## 5.1 EXPERIMENTAL RESULTS

Results on the four benchmark datasets are provided in Table 1, Table 2, Table 6, and Table 7 in the appendix. We observe that LDC-MTL outperforms most existing methods on both the CelebA and QM9 datasets, achieving almost the lowest performance drops of $\Delta m\% = -1.31$ and $\Delta m\% = 49.5$, except for ConsMTL. Detailed results for the QM9 dataset illustrate that it achieves a balanced performance across all tasks. These results highlight the effectiveness of our method in handling a large number of tasks in both classification and regression settings. Meanwhile, it achieves almost the lowest performance drop, with $\Delta m\% = -0.57$ on the Cityscapes dataset, while delivering comparable results on the NYU-v2 dataset, where the detailed results are shown in Table 7 in the appendix. These findings highlight the capability of LDC-MTL to handle mixed MTL scenarios.

**Efficiency comparison.** We compare the running time of well-performing approaches in Figure 4. In particular, our method introduces negligible overhead compared to LS with at most a $1.11\times$ increase, aligning with other $\mathcal{O}(1)$ methods such as GO4Align and FAMO. In contrast, gradient manipulation methods, which take the computational cost $\mathcal{O}(K)$, become significantly slower in many-task scenarios. For example, Nash-MTL requires approximately $12\times$, and the state-of-the-art ConsMTL requires $11\times$ more training time than LDC-MTL on the CelebA dataset.

Table 3: Statistics of loss values for LDC-MTL, GO4Align and LS on the CelebA dataset. Lower mean and std indicate better and more stable performance.

| Method | Mean ↓ | Std ↓ | Min | Max |
|---|---|---|---|---|
| LDC-MTL | **0.189** | **0.133** | 0.029 | **0.538** |
| GO4Align | 0.238 | 0.154 | **0.026** | 0.660 |
| FAMO | 0.231 | 0.141 | 0.027 | 0.621 |
| LS | 0.287 | 0.195 | 0.032 | 0.729 |

## 5.2 EXPERIMENTAL ANALYSIS

**Loss discrepancy and gradient conflict.** To demonstrate the effectiveness of our bilevel formulation for loss discrepancy control, we conduct a detailed analysis of the loss distribution on the CelebA dataset, comparing LS and GO4Align with our proposed method. As shown in Figure 8 and statistics in Table 3, the distribution of all 40 task-specific losses reveals that our approach yields more concentrated and consistently lower values. Moreover, we randomly select 8 out of 40 tasks and check the gradient cosine similarity among them. Figure 9 illustrates the cosine similarities of task gradients after the 15th epoch, which shows that the gradient conflict is mitigated as a side-effect.

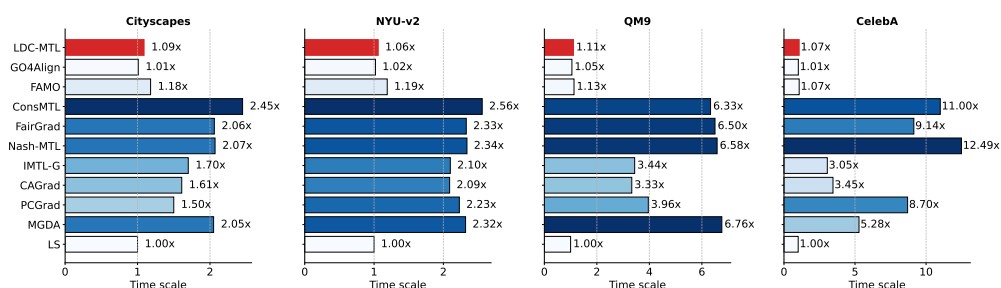

Figure 4: Time scale comparison among well-performing approaches, with LS considered the reference method for standard time.

**Pareto front Comparison.** Prior work has noted that certain multi-task learning methods, such as RLW and PCGrad, do not outperform LS, which can approximate the Pareto front through weight sweeping (Kurin et al., 2022; Xin et al., 2022). To this end, we conduct a careful comparison among our LDC-MTL, GO4Align, FAMO and LS with weight sweeping on the Cityscapes dataset. From Table 9 in the appendix, even after careful weight sweeping, LS does not outperform our approach. The scatter plot in Figure 5 (a-b) also illustrates that LDC-MTL solutions form the Pareto front.

**Impact of task weight.** To further investigate the importance of dynamic weighting, we provide the evolution of task weights for the 11 tasks on the QM9 dataset in Figure 5 (c). It is evident that the task weights adapt meaningfully during the early stages of training and gradually converge later on. Besides, the converged weights are nearly equal, recovering to LS. However, LS does not perform well as shown in Table 6. These findings suggest that, for dynamic-weight methods, the *training trajectory*, not just the final weights, plays a critical role in achieving strong performance.

Table 4: Parameter tuning results on the CelebA dataset.

| Method | $\Delta m\% \downarrow$ |
|---|---|
| FairGrad (Ban & Ji, 2024) | 0.37 |
| LDC-MTL ($\lambda = 0.005$) | -1.27 |
| LDC-MTL ($\lambda = 0.008$) | -1.16 |
| LDC-MTL ($\lambda = 0.01$) | **-1.31** |
| LDC-MTL ($\lambda = 0.02$) | -0.96 |

**Hyperparameter sensitivity.** In our method, hyperparameters include the step size $\alpha$ and the penalty constant $\lambda$. For the step size, we adopt the settings from prior experiments without extensive tuning. While we use the same step size for updates to both $W$ and $x$ in our implementation, these can be adjusted independently in practice. For the penalty constant $\lambda$, we determine optimal values through a grid search and provide additional experimental results in Table 4 and Table 8 in the appendix.

## 6 THEORETICAL ANALYSIS

In this section, we provide convergence analysis for our LDC-MTL method.

**Definition 1** *Given $L > 0$, a function $\ell$ is said to be L-Lipschitz-continuous on $\mathcal{X}$ if it holds for any $x, x' \in \mathcal{X}$ that $\|\ell(x) - \ell(x')\| \leq L\|x - x'\|$. A function $\ell$ is said to be L-Lipschitz-smooth if its gradient is L-Lipschitz-continuous.*

**Definition 2 (Pareto stationarity)** *We say $x$ is an $\epsilon$-accurate Pareto stationary point for loss functions $\{l_i(x)\}$ if $\min_{w \in \mathcal{W}} \|G(x)w\|^2 = \mathcal{O}(\epsilon)$, where $G(x) = [\nabla l_1(x), \nabla l_2(x), ..., \nabla l_K(x)]^\top$.*

Inspired by Shen & Chen (2023), we also define the following two surrogates of the original bilevel problem in eq. (1).

**Definition 3** *Define two surrogate bilevel problems as*

$$\mathcal{BP}_\lambda : \min_{W,x} f(W, x) + \lambda(g(W, x) - g(W, x^*)), \mathcal{BP}_\epsilon : \min_{W,x} f(W, x) \ \ s.t. \ g(W, x) - g(W, x^*) \leq \epsilon,$$

*where $\mathcal{BP}_\lambda$ is the penalized bilevel problem, and $\mathcal{BP}_\epsilon$ recovers to the original problem if $\epsilon = 0$.*

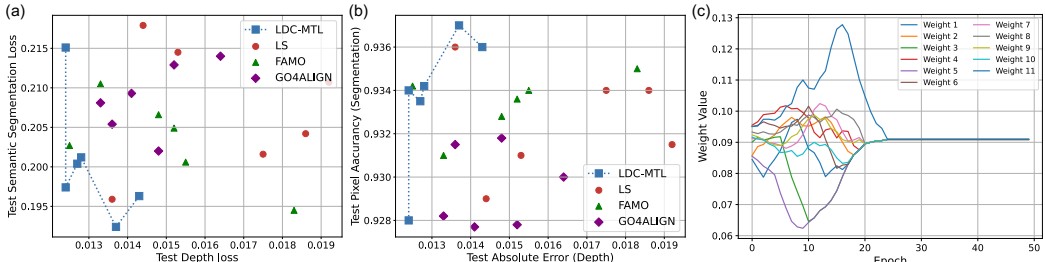

Figure 5: (a–b) Comparison between LDC-MTL, weight-swept LS, FAMO, and GO4Align with different random seeds on the Cityscapes dataset. In both cases, LDC-MTL solutions lie on the Pareto frontier. (c) Weight change of 11 tasks using LDC-MTL during the training on the QM9 dataset.

**Assumption 1 (Lipschitz and smoothness)** *There exists a constant $L$ such that the upper-level function $f(W, \cdot)$ is $L$-Lipschitz continuous. There exists constants $L_f$ and $L_g$ such that functions $f(W, x)$ and $g(W, x)$ are $L_f$- and $L_g$-Lipschitz-smooth.*

**Assumption 2 (Polyak-Lojasiewicz (PL) condition)** *The lower-level function $g(W, \cdot)$ satisfies the $\frac{1}{\mu}$-PL condition if there exists a $\mu > 0$ such that given any $W$, it holds for any feasible $x$ that,*

$$\|\nabla_x g(W, x)\|^2 \geq \frac{1}{\mu}(g(W, x) - g(W, x^*)).$$

Lipschitz continuity and smoothness are standard assumptions in the study of bilevel optimization (Ghadimi & Wang, 2018; Ji et al., 2021). While the absolute values in the upper-level function in eq. (1) are non-smooth, they can be easily modified to ensure smoothness, such as by using a soft absolute value function of the form $y = \sqrt{x^2 + \gamma}$ where $\gamma$ is a small positive constant. Moreover, the PL condition can be satisfied in over-parameterized neural network settings (Mei et al., 2020; Frei & Gu, 2021). The following theorem presents the convergence analysis of our algorithms.

**Theorem 1** *Suppose Assumptions 1- 2 are satisfied. Select hyperparameters*

$$\alpha \in \left(0, \frac{1}{L_f + \lambda(2L_g + L_g^2\mu)}\right], \ \beta \in (0, \frac{1}{L_g}], \lambda = L\sqrt{3\mu\epsilon^{-1}}, \ and \ N = \Omega(\log(\alpha t)).$$

*(i) Our method with the updates eq. (3) and eq. (4) (i.e., Algorithm 2 in the appendix) finds an $\epsilon$-accurate stationary point of the problem $\mathcal{BP}_\lambda$. If this stationary point is a local/global solution to $\mathcal{BP}_\lambda$, it is also a local/global solution to $\mathcal{BP}_\epsilon$. Furthermore, it is also an $\epsilon$-accurate Pareto stationary point for loss functions $l_i(x), i = 1, ..., K$.*

*(ii) Moreover, if $\|\nabla_W g(W^t, z_N^t)\| = \mathcal{O}(\epsilon)$ for $t = 1, ..., T$. The simplified method in Algorithm 1 also achieves the same convergence guarantee as that in $(i)$.*

The complete proof is provided in Theorem 2. In the first part of Theorem 1, we establish a connection between the stationarity of $\mathcal{BP}_\lambda$ and Pareto stationarity. Secondly, it introduces an additional gradient vanishing assumption, which has been validated in our experiments. It demonstrates that our simplified LDC-MTL method can also attain an $\epsilon$-accurate stationary point for the problem $\mathcal{BP}_\lambda$ and an $\epsilon$-accurate Pareto stationary point for the original loss functions.

## 7 CONCLUSION

We introduced LDC-MTL, a scalable loss discrepancy control approach for multi-task learning based on bilevel optimization. Our method achieves efficient loss discrepancy control with only $\mathcal{O}(1)$ time and memory complexity while guaranteeing convergence to both a stationary point of the bilevel problem and an $\epsilon$-accurate Pareto stationary point for all task loss functions. Extensive experiments demonstrate that LDC-MTL outperforms existing methods in both accuracy and efficiency, highlighting its effectiveness for large-scale MTL. For future work, we plan to explore the application of our method to broader multi-task learning problems, including recommendation systems.

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

# A EXPERIMENT DETAILS

## A.1 EXPERIMENTAL SETUP

**Image-Level Classification.** CelebA (Liu et al., 2015), one of the most widely used datasets, is a large-scale facial attribute dataset containing over 200K celebrity images. Each image is annotated with 40 attributes, such as the presence of eyeglasses and smiling. Following the experimental setup in Ban & Ji (2024), we treat CelebA as a 40-task multi-task learning (MTL) classification problem, where each task predicts the presence of a specific attribute. Since all tasks involve binary classification with the same *binary cross-entropy* loss function, we do not apply any normalization for both options of $\tau = \mathbf{1}$ and $\tau = \sigma$. The network architecture consists of a 9-layer convolutional neural network (CNN) as the shared model, with multiple linear layers serving as task-specific heads. We train the model for 15 epochs using the Adam optimizer with a batch size of 256.

**Regression.** QM9 (Ramakrishnan et al., 2014) dataset is another widely used benchmark for multi-task regression problems in quantum chemistry. It contains 130K molecules represented as graphs, and 11 properties to be predicted. Though all tasks share the same loss function, *mean squared error*, they exhibit significantly varying scales: a phenomenon commonly observed in regression tasks but less prevalent in classification tasks, as shown in Figure 6. To mitigate this scale discrepancy, we first adopt the logarithmic normalization such that $\tilde{l}_i = \log\left(\frac{l_i}{l_{i,0}}\right)$, where $l_{i,0}$ represents the initial loss value for the $i$-th task at each epoch, motivated by Liu et al. (2024), for both options of $\tau = \mathbf{1}$ and $\tau = \sigma$. Our experiments demonstrate that this initialization approach stabilizes training by reducing large fluctuations caused by significant scale variations. Following the experimental setup in Liu et al. (2024); Navon et al. (2022), we use the same model and data split, 110K molecules for training, 10k for validation, and the rest 10k for testing. The model is trained for 300 epochs with a batch size of 120. The learning rate starts at 1e-3 and is reduced whenever the validation performance stagnates for 5 consecutive epochs.

**Dense Prediction.** The Cityscapes dataset (Cordts et al., 2016) consists of 5000 street-scene images designed for two tasks: 7-class semantic segmentation (a classification task) and depth estimation (a regression task). Similarly, the NYU-v2 dataset (Silberman et al., 2012) is widely used for indoor scene understanding and contains 1449 densely annotated images. It includes one pixel-level classification task, semantic segmentation, and two pixel-level regression tasks, 13-class depth estimation, plus surface normal prediction. These datasets provide benchmarks for evaluating the performance of our method in mixed multi-task settings. Since the number of tasks is small and the loss values exhibit minimal variation, we applied rescaled normalization when selecting $\tau = \sigma$ and no normalization when selecting $\tau = \mathbf{1}$. The rescaled normalization normalizes loss values by rescaling each task's loss using its initial loss value $l'_i$, such that $\tilde{l}_i = \frac{l_i}{l'_i}$. The resulting normalized loss reflects the training progress and ensures comparability across tasks. We follow the same experimental setup described in Liu et al. (2021a); Navon et al. (2022) and adopt MTAN (Liu et al., 2019) as the backbone, which incorporates task-specific attention modules into SegNet (Badrinarayanan et al., 2017). Both models are trained for 200 epochs, with batch sizes of 8 for Cityscapes and 2 for NYU-v2. The learning rates are initialized at 3e-4 and 1e-4 for the first 100 epochs and reduced by half for the remaining epochs, respectively. In a word, the hyperparameter choices are summarized in Table 5.

Table 5: Training hyperparameters combination and the best results per dataset.

| Dataset | Stepsize | Penalty constant | $\tau$ | $\Delta m\%$ |
|---|---|---|---|---|
| CelebA | 1e-03 | 0.01 | $\sigma(W)$ | -1.31±0.26 |
| QM9 | 1e-03 | 0.05 | $\sigma(W)$ | 49.5±3.64 |
| Cityscapes | 3e-04 | 0.1 | 1 | -0.57±1.17 |
| NYU-v2 | 8e-05 | 0.05 | 1 | -4.40±0.74 |

**Router model details.** In general, $\sigma(W)$ refers to a lightweight routing network parameterized by $W$, which takes as input the output features from the shared backbone of the model. While the router design varies slightly across the four datasets, it remains consistently simple and small in scale relative to the full model. Below, we provide the full details for all four datasets:

Cityscapes: A 3-layer MLP maps 128-dimensional input features to 2 output weights, with a total of 10,402 parameters. The full SegNet model contains 41,217,386 parameters. NYU-v2: A 3-layer MLP maps 192-dimensional input features to 3 output weights, comprising 23,184 parameters. The SegNet model used here contains 44,229,652 parameters. QM9: A directly trainable weight vector of shape [11, 1] is used, corresponding to 11 tasks. The model contains 617,686 parameters in total. CelebA: A directly trainable weight vector of shape [40, 1] is used for the 40 classification tasks. The full model has 5,217,720 parameters. Importantly, $\sigma(W)$ is only used during the training phase for dynamic weighting. During inference, the model follows the same setup as prior methods, such as FairGrad and FAMO.

## A.2 LOSS AND GRADIENT SCALES

We show that loss can vary significantly in scale. For example, as shown in Figure 6, we observed that the loss values during training across 11 regression tasks in the QM9 dataset exhibit substantial differences in magnitude, with loss ratios exceeding 1000 in certain instances. Moreover, we have also drawn the loss progression over time of LS. Tasks 1–4 are easier to optimize and dominate the training when using linear scalarization, leading other tasks to converge suboptimally. In contrast, LDC-MTL accounts for loss scale and adaptively reweights tasks using bilevel optimization, enabling better balance. For example, tasks 6–10 improve their final loss scale from $10^{-3}$ to $10^{-5}$.

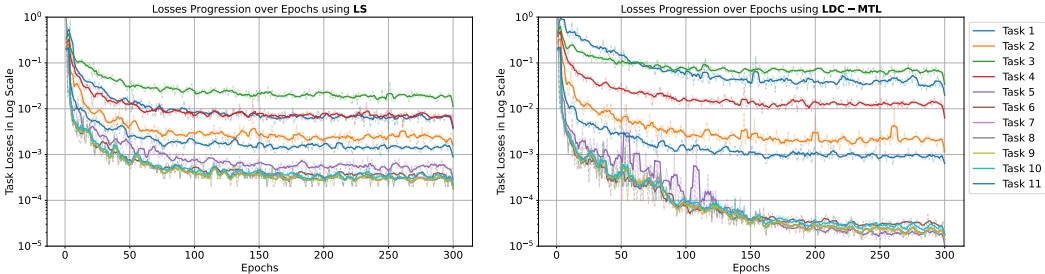

Figure 6: Curves of loss values during the training process for all 11 tasks on the QM9 dataset using different methods. The loss values vary significantly across different tasks. For LS, some tasks converge suboptimally due to the loss discrepancy.

Moreover, our experiments reveal that the gradient norm $\|\nabla_W g(W^t, z_N^t)\|$ remains sufficiently small, typically orders of magnitude smaller than the gradient norm $\|\nabla_W g(W^t, x^t)\|$, which is used to update outer parameters $W$. This behavior is illustrated in Figure 7 for both CelebA and Cityscapes datasets. Specifically, we set $N = 50$ during training. In average, the ratio $\|\nabla_W g(W^t, x^t)\|/\|\nabla_W g(W^t, z_N^t)\|$ exceeds 100, despite some fluctuations.

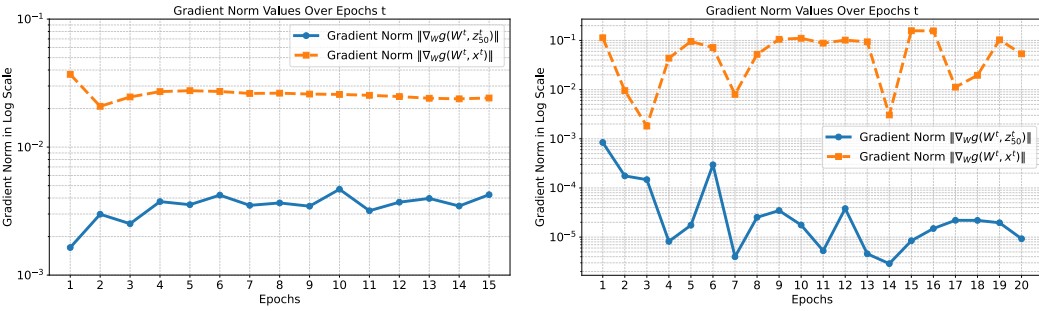

Figure 7: Gradient norm values during the training process on the CelebA and Cityscapes datasets.

## A.3 TOY EXAMPLE

To better understand the benefits of our method, we illustrate the training trajectory along with the training time in a toy example of 2-task learning following the same setting in FAMO (Liu et al., 2024). The loss functions $L_1(x), L_2(x)$, where $x$ is the model parameter, of two tasks are listed below.

$$L_1(x) = 0.1 \times (c_1(x)f_1(x) + c_2(x)g_1(x)), \quad L_2(x) = c_1(x)f_2(x) + c_2(x)g_2(x) \quad \text{where}$$

$$f_1(x) = \log\big(\max(|0.5(-x_1 - 7) - \tanh(-x_2)|, \ 0.000005)\big) + 6,$$

$$f_2(x) = \log\big(\max(|0.5(-x_1 + 3) - \tanh(-x_2) + 2|, \ 0.000005)\big) + 6,$$

$$g_1(x) = \big((-x_1 + 7)^2 + 0.1 * (-x_2 - 8)^2\big)/10 - 20,$$

$$g_2(x) = \big((-x_1 - 7)^2 + 0.1 * (-x_2 - 8)^2\big)/10 - 20,$$

$$c_1(x) = \max(\tanh(0.5 * x_2), \ 0) \quad \text{and} \quad c_2(x) = \max(\tanh(-0.5 * x_2), \ 0). \quad (5)$$

In Figure 1, the black dots represent 5 chosen initial points $\{(-8.5, 7.5), (-8.5, 5), (0, 0), (9, 9), (10, -8)\}$ while the black stars represent the converging points on the Pareto front. We use the Adam optimizer and train each method for 50k steps. Our method can always converge to balanced results efficiently. We use Adam optimizer with a learning rate of 1e-3. The training time is recalculated according to real-time ratios in our machine. We find that LS and MGDA do not converge to balanced points, while FAMO converges to balanced results to some extent. Meanwhile, our method with rescale normalization can always converge to balanced results efficiently.

## A.4 DETAILED RESULTS

Here we provide detailed results of NYU-v2 in Table 7 and QM9 in Table 6.

Table 6: Detailed results of on QM9 (11-task) dataset. Each experiment is repeated 3 times, and the average is reported. The best results are highlighted in **bold**, while the second-best results are indicated with underlines.

| METHOD | $\mu$ | $\alpha$ | $\epsilon_{HOMO}$ | $\epsilon_{LUMO}$ | $\langle R^2\rangle$ | ZPVE | $U_0$ | $U$ | $H$ | $G$ | $c_v$ | MR↓ | $\Delta m\%\downarrow$ |
|---|---|---|---|---|---|---|---|---|---|---|---|---|---|
| | | | | | | MAE↓ | | | | | | | |
| STL | 0.067 | 0.181 | 60.57 | 53.91 | 0.502 | 4.53 | 58.8 | 64.2 | 63.8 | 66.2 | 0.072 | | |
| LS | 0.106 | 0.325 | **73.57** | 89.67 | 5.19 | 14.06 | 143.4 | 144.2 | 144.6 | 140.3 | 0.128 | 11.27 | 177.6 |
| SI | 0.309 | 0.345 | 149.8 | 135.7 | 1.00 | 4.50 | 55.3 | 55.75 | 55.82 | 55.27 | 0.112 | 8.09 | 77.8 |
| RLW (LIN ET AL., 2021) | 0.113 | 0.340 | 76.95 | 92.76 | 5.86 | 15.46 | 156.3 | 157.1 | 157.6 | 153.0 | 0.137 | 12.73 | 203.8 |
| DWA (LIU ET AL., 2019) | 0.107 | 0.325 | 74.06 | 90.61 | 5.09 | 13.99 | 142.3 | 143.0 | 143.4 | 139.3 | 0.125 | 12.73 | 175.3 |
| UW (KENDALL ET AL., 2018) | 0.386 | 0.425 | 166.2 | 155.8 | 1.06 | 4.99 | 66.4 | 66.78 | 66.80 | 66.24 | 0.122 | 10.95 | 108.0 |
| FAMO (LIU ET AL., 2024) | 0.15 | 0.30 | 94.0 | 95.2 | 1.63 | 4.95 | 70.82 | 71.2 | 71.2 | 70.3 | 0.10 | 7.64 | 58.5 |
| GO4ALIGN (SHEN ET AL., 2024B) | 0.17 | 0.35 | 102.4 | 119.0 | 1.22 | 4.94 | 53.9 | 54.3 | 54.3 | 53.9 | 0.11 | 7.41 | 52.7 |
| STCH (LIN ET AL., 2024) | 0.166 | 0.260 | 94.48 | 101.2 | 1.850 | 4.88 | 58.34 | 58.68 | 58.70 | 58.27 | 0.104 | 6.82 | 56.9 |
| MGDA (DÉSIDÉRI, 2012) | 0.217 | 0.368 | 126.8 | 104.6 | 3.22 | 5.69 | 88.37 | 89.4 | 89.32 | 88.01 | 0.120 | 11.68 | 120.5 |
| PCGRAD (YU ET AL., 2020) | 0.106 | 0.293 | 75.85 | 88.33 | 3.94 | 9.15 | 116.36 | 116.8 | 117.2 | 114.5 | 0.110 | 9.36 | 125.7 |
| CAGRAD (LIU ET AL., 2021A) | 0.118 | 0.321 | 83.51 | 94.81 | 3.21 | 6.93 | 113.99 | 114.3 | 114.5 | 112.3 | 0.116 | 10.45 | 112.8 |
| IMTL-G (LIU ET AL., 2021B) | 0.136 | 0.287 | 98.31 | 93.96 | 1.75 | 5.69 | 101.4 | 102.4 | 102.0 | 100.1 | 0.096 | 8.95 | 77.2 |
| NASH-MTL (NAVON ET AL., 2022) | **0.102** | **0.248** | 82.95 | **81.89** | 2.42 | 5.38 | 74.5 | 75.02 | 75.10 | 74.16 | **0.093** | 6.18 | 62.0 |
| FAIRGRAD (BAN & JI, 2024) | 0.117 | 0.253 | 87.57 | 84.00 | 2.15 | 5.07 | 70.89 | 71.17 | 71.21 | 70.88 | 0.095 | 6.55 | 57.9 |
| CONSMTL (QIN ET AL., 2025)[1] | 0.115 | 0.202 | 82.69 | 67.58 | 1.61 | 3.33 | 48.84 | 49.04 | 49.07 | 49.63 | 0.077 | 2.55 | 23.2 |
| LDC-MTL | 0.23 | 0.29 | 123.89 | 111.95 | **0.97** | 3.99 | 42.73 | 43.1 | 43.2 | 43.1 | 0.097 | 5.27 | **49.5±3.64** |

## A.5 PARAMETER TUNING

In our experiments, the penalty constant $\lambda$ requires some tuning effort, whereas the choice of step size had relatively less impact and did not require extensive tuning. We have included additional results on the CelebA and Cityscapes datasets in Table 4 and Table 8 that explore the effect of varying $\lambda$, and found that values in the range $\lambda \in [0.02, 0.1]$ consistently yield strong performance. Besides, we use $\Delta m\%$ as the evaluation metric, which aggregates performance across all tasks based on their definitions. As a result, it may exhibit slightly higher variance. This behavior is consistent with other experiments; for example, similar variance patterns can be observed in Table 5 in Xiao et al. (2024).

---

[1]ConsMTL is a concurrent state-of-the-art gradient-manipulation method that formulates MTL as a bilevel optimization over shared and task-specific parameters

Table 7: Results on NYU-v2 (3-task) dataset. Each experiment is repeated 3 times with different random seeds, and the average is reported.

| METHOD | SEGMENTATION | | DEPTH | | SURFACE NORMAL | | | | | MR↓ | $\Delta m\%$↓ |
|---|---|---|---|---|---|---|---|---|---|---|---|
| | | | | | ANGLE DISTANCE ↓ | | WITHIN $t°$ ↑ | | | | |
| | MIOU ↑ | PIX ACC ↑ | ABS ERR ↓ | REL ERR ↓ | MEAN | MEDIAN | 11.25 | 22.5 | 30 | | |
| STL | 38.30 | 63.76 | 0.6754 | 0.2780 | 25.01 | 19.21 | 30.14 | 57.20 | 69.15 | | |
| LS | 39.29 | 65.33 | 0.5493 | 0.2263 | 28.15 | 23.96 | 22.09 | 47.50 | 61.08 | 14.44 | 5.59 |
| SI | 38.45 | 64.27 | 0.5354 | 0.2201 | 27.60 | 23.37 | 22.53 | 48.57 | 62.32 | 13.00 | 4.39 |
| RLW (LIN ET AL., 2021) | 37.17 | 63.77 | 0.5759 | 0.2410 | 28.27 | 24.18 | 22.26 | 47.05 | 60.62 | 17.22 | 7.78 |
| DWA (LIU ET AL., 2019) | 39.11 | 65.31 | 0.5510 | 0.2285 | 27.61 | 23.18 | 24.17 | 50.18 | 62.39 | 13.44 | 3.57 |
| UW (KENDALL ET AL., 2018) | 36.87 | 63.17 | 0.5446 | 0.2260 | 27.04 | 22.61 | 23.54 | 49.05 | 63.65 | 13.00 | 4.05 |
| FAMO (LIU ET AL., 2024) | 38.88 | 64.90 | 0.5474 | 0.2194 | 25.06 | 19.57 | 29.21 | 56.61 | 68.98 | 7.44 | -4.10 |
| GO4ALIGN (SHEN ET AL., 2024B) | **40.42** | 65.37 | 0.5492 | 0.2167 | 24.76 | **18.94** | 30.54 | 57.87 | 69.84 | 4.11 | **-6.08** |
| STCH (LIN ET AL., 2024) | 41.35 | 66.07 | 0.4965 | 0.2010 | 26.55 | 21.81 | 24.84 | 51.39 | 64.86 | 6.28 | -1.35 |
| MGDA (DÉSIDÉRI, 2012) | 30.47 | 59.90 | 0.6070 | 0.2555 | 24.88 | 19.45 | 29.18 | 56.88 | 69.36 | 10.56 | 1.38 |
| PCGRAD (YU ET AL., 2020) | 38.06 | 64.64 | 0.5550 | 0.2325 | 27.41 | 22.80 | 23.86 | 49.83 | 63.14 | 13.78 | 3.97 |
| GRADDROP (CHEN ET AL., 2020) | 39.39 | 65.12 | 0.5455 | 0.2279 | 27.48 | 22.96 | 23.38 | 49.44 | 62.87 | 12.56 | 3.58 |
| CAGRAD (LIU ET AL., 2021A) | 39.79 | 65.49 | 0.5486 | 0.2250 | 26.31 | 21.58 | 25.61 | 52.36 | 65.58 | 8.61 | 0.20 |
| IMTL-G (LIU ET AL., 2021B) | 39.35 | 65.60 | 0.5426 | 0.2256 | 26.02 | 21.19 | 26.20 | 53.13 | 66.24 | 7.89 | -0.76 |
| MOCO (FERNANDO ET AL., 2023) | 40.30 | **66.07** | 0.5575 | **0.2135** | 26.67 | 21.83 | 25.61 | 51.78 | 64.85 | 8.44 | 0.16 |
| NASH-MTL (NAVON ET AL., 2022) | 40.13 | 65.93 | **0.5261** | 0.2171 | 25.26 | 20.08 | 28.40 | 55.47 | 68.15 | 5.67 | -4.04 |
| FAIRGRAD (BAN & JI, 2024) | 39.74 | 66.01 | 0.5377 | 0.2236 | 24.84 | 19.60 | 29.26 | 56.58 | 69.16 | 5.22 | -4.66 |
| CONSMTL (QIN ET AL., 2025) | 40.33 | 65.32 | 0.5491 | 0.2151 | 24.35 | 18.80 | 31.06 | 58.28 | 70.31 | 3.56 | -6.72 |
| LDC-MTL | 38.04 | 65.92 | 0.5402 | 0.2278 | **24.70** | 19.19 | 29.97 | 57.44 | 69.69 | 5.78 | -4.40±0.74 |

Table 8: Additional results on Cityscapes (2-task) dataset with different $\lambda$ values.

| METHOD | SEGMENTATION | | DEPTH | | $\Delta m\%$ ↓ |
|---|---|---|---|---|---|
| | MIOU ↑ | PIX ACC ↑ | ABS ERR ↓ | REL ERR ↓ | |
| STL | 74.01 | 93.16 | 0.0125 | 27.77 | |
| FAIRGRAD (BAN & JI, 2024) | 75.72 | 93.68 | 0.0134 | 32.25 | 5.18 |
| LDC-MTL ($\tau = 1, \lambda = 0.02$) | 73.18 | 92.78 | 0.0124 | 29.67 | **1.96**±1.25 |
| LDC-MTL ($\tau = 1, \lambda = 0.05$) | 74.50 | 93.40 | 0.0124 | 28.99 | **0.79**±1.10 |
| LDC-MTL ($\tau = 1, \lambda = 0.06$) | 74.84 | 93.43 | 0.0123 | 29.61 | **0.92**±0.95 |
| LDC-MTL ($\tau = 1, \lambda = 0.07$) | 75.40 | 93.42 | 0.0125 | 29.25 | **0.88**±1.01 |
| LDC-MTL ($\tau = 1, \lambda = 0.08$) | 75.34 | 93.35 | 0.0127 | 29.70 | **1.64**±1.04 |
| LDC-MTL ($\tau = 1, \lambda = 0.09$) | 74.97 | 93.50 | 0.0123 | 28.90 | **0.18**±0.96 |
| LDC-MTL ($\tau = 1, \lambda = 0.1$) | 74.53 | 93.42 | 0.0128 | 26.79 | **-0.57**±1.17 |

## A.6 LOSS DISCREPANCY AND GRADIENT CONFLICT

To demonstrate the effectiveness of our bilevel formulation for loss discrepancy control, we conduct a detailed analysis of the loss distribution on the CelebA dataset, comparing linear scalarization (LS) and GO4Align with our proposed method. As shown in Figure 8 and statistics in Table 3, the distribution of all 40 task-specific losses reveals that our approach yields more concentrated and consistently lower values.

Except for the loss discrepancy, we randomly select 8 out of 40 tasks and have checked the gradient cosine similarity among tasks on the CelebA dataset. Figure 9 illustrates the cosine similarities of task gradients after the 15th epoch, which shows that the gradient conflict is mitigated.

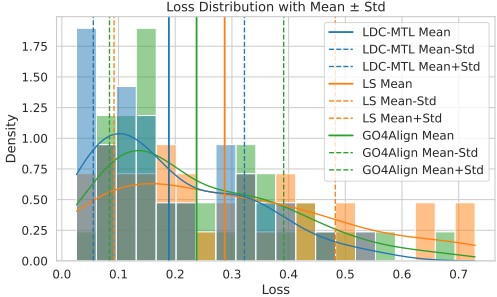

Figure 8: Comparison of loss distributions for LDC-MTL, GO4Align and LS on the CelebA dataset.

## A.7 COMPARISON WITH WEIGHT-SWEPT LS

We have operated a weight sweep on linear scalarization, and the result is shown in Table 9. From this table, we find that even after a careful weight sweep, LS does not perform better than our method.

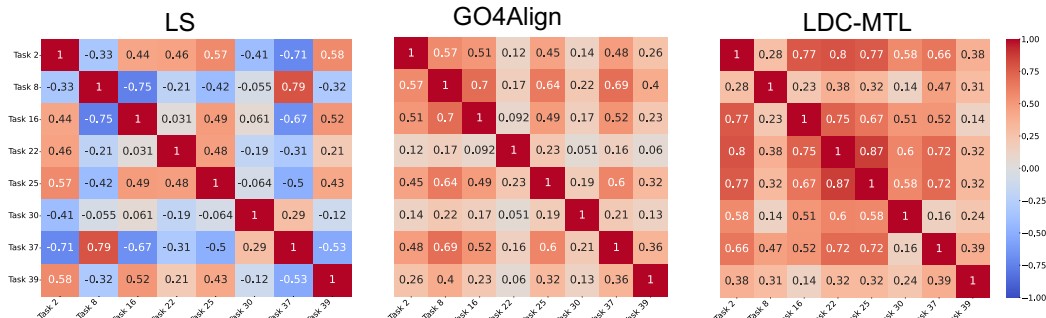

Figure 9: Cosine similarities of task gradients for LS, GO4Align and LDC-MTL, respectively; LDC-MTL exhibits much higher gradient similarity among tasks, suggesting reduced gradient conflict.

Moreover, we have also provided a scatter plot in Figure 5 (a-b) using results in Table 8 and Table 9 following Figure 8 in Xin et al. (2022). In both cases, our method forms the Pareto frontier.

Table 9: Additional results on Cityscapes (2-task) dataset with different weights of LS.

| METHOD | SEGMENTATION | | DEPTH | | $\Delta m\% \downarrow$ |
|---|---|---|---|---|---|
| | MIOU $\uparrow$ | PIX ACC $\uparrow$ | ABS ERR $\downarrow$ | REL ERR $\downarrow$ | |
| STL | 74.01 | 93.16 | 0.0125 | 27.77 | |
| LDC-MTL ($\tau = 1, \lambda = 0.1$) | 74.53 | 93.42 | 0.0128 | 26.79 | **-0.57**±1.17 |
| LS ($w_1 = 0.1, w_2 = 0.9$) | 74.00 | 92.92 | 0.0144 | 29.23 | 5.13±1.10 |
| LS ($w_1 = 0.2, w_2 = 0.8$) | 75.09 | 93.58 | 0.0136 | 33.73 | 7.18±0.99 |
| LS ($w_1 = 0.3, w_2 = 0.7$) | 74.10 | 93.08 | 0.0153 | 35.38 | 12.38±1.78 |
| LS ($w_1 = 0.4, w_2 = 0.6$) | 74.95 | 93.43 | 0.0175 | 43.15 | 23.47±1.77 |
| LS ($w_1 = 0.6, w_2 = 0.4$) | 74.48 | 93.39 | 0.0186 | 43.93 | 26.53±1.77 |
| LS ($w_1 = 0.8, w_2 = 0.2$) | 74.24 | 93.15 | 0.0192 | 61.24 | 43.19±6.50 |

## A.8 ABLATION STUDY

**Task orders.** To investigate the impact of loss ordering in the upper-level function $f(W, x^*)$ in eq. (1), we randomly shuffled the order of loss values before computing the (weighted) loss gaps on the CelebA dataset. The results are reported in Table 10 where CelebA(reorder) represents random reordering on task losses before computing $f(W, x)$. Our findings indicate that reordering the losses does not lead to significant performance differences. This suggests that the effect of task ordering is minimal, likely because the loss values are already on a comparable scale.

Table 10: Additional result on the loss orders with CelebA.

| Dataset | $\Delta m\%$ |
|---|---|
| CelebA | -1.31 |
| CelebA(reorder), seed=0 | -1.30 |
| CelebA(reorder), seed=1 | -1.00 |
| CelebA(reorder), seed=2 | -1.62 |

## B ADDITIONAL INFORMATION

**Additional algorithms.** Here, we present the complete version of the double-loop algorithm for solving the penalized bilevel problem, $\mathcal{BP}_\lambda$ in eq. (2), as detailed in Algorithm 2. Notably, the local

or global solution of $\mathcal{BP}_\lambda$ obtained by Algorithm 2 also serves as a local or global solution to $\mathcal{BP}_\epsilon$, as established by Proposition 2 in Shen & Chen (2023).

**Additional discussion with related works.** The previous works, AO4Align (Shen et al., 2024b) and ConsMTL (Qin et al., 2025) have bilevel optimization formulation, but there are fundamental differences.

*Differences with GO4Align*: In GO4Align, the lower-level problem is designed to identify **task groupings**, while the upper-level objective minimizes the weighted task loss. It updates the group assignment matrix implicitly within the lower-level optimization before updating model parameters. Lastly, our method provides a convergence analysis achieving a Pareto stationary point, whereas GO4Align does not offer such theoretical guarantees.

*Differences with ConsMTL:* ConsMTL (Qin et al., 2025) addresses gradient conflict by manipulating per-task gradients, with the upper level aggregating gradients and the lower level updating task-specific parameters. This leads to an $\mathcal{O}(K)$ overhead and lacks convergence guarantees to Pareto-stationary points. Our method, by contrast, maintains $\mathcal{O}(1)$ complexity and provides such guarantees.

---

**Algorithm 2: Double-loop First-order Method**

**Initialize:** $W^0, x^0, z_0^0$
**for** $t = 0, 1, \ldots, T-1$ **do**
    Warm start: $z_0^t = x^t$
    **for** $n = 0, 1, \ldots, N$ **do**
        $z_{n+1}^t = z_n^t - \beta\lambda\nabla_z g(W^t, z_n^t)$
    **end for**
    $x^{t+1} = x^t - \alpha\left(\nabla_x f(W^t, x^t) + \lambda\nabla_x g(W^t, x^t)\right)$
    $W^{t+1} = W^t - \alpha\left(\nabla_W f(W^t, x^t) + \lambda\left(\nabla_W g(W^t, x^t) - \nabla_W g(W^t, z_N^t)\right)\right)$
**end for**

---

# C ANALYSIS

## C.1 COMPARISON WITH PENALTY-BASED BILEVEL OPTIMIZATION METHODS

While our method adopts a penalty-based bi-level optimization approach, our Algorithm 1 has adjustments based on experimental exploration. First, as discussed in Section 4.3, both Kwon et al. (2023); Shen & Chen (2023) require a sub-loop to approximate the lower-level optimum $x_t^*$. In contrast, our empirical analysis in Figure 7 shows that this approximation can be safely eliminated, allowing Algorithm 1 to operate as a fully single-loop method.

In terms of the analysis, the lower-level problem in Kwon et al. (2023) has to be assumed to be strongly convex, while we assume a relaxed PL condition. Meanwhile, although Shen & Chen (2023) considers the PL condition, its analysis is under a single-task setting, and there is no analysis of whether the penalty-based method converges to Pareto stationary points in MOO problems. Therefore, in our theorem, we establish a connection between the stationarity of $\mathcal{BP}_\lambda$ and Pareto stationarity. Moreover, with a gradient-vanishing assumption which is validated in our experiment, our method can also attain an $\epsilon$-accurate stationary point for the problem $\mathcal{BP}_\lambda$ and an $\epsilon$-accurate Pareto stationary point for the original loss functions.

## C.2 PROOF

In the analysis, we need the following definitions.

$$x_t^* = \arg\min_x g(W^t, x),$$

$$G(x) = [\nabla l_1(x), \nabla l_2(x), ..., \nabla l_K(x)]$$

$$F(\theta^t) = f(\theta^t) + \lambda p(\theta^t), \Phi(\theta^t) = f(\theta^t) + \lambda g(\theta^t),$$

$$\text{where } \theta^t = (W^t, x^t), p(\theta^t) = g(W^t, x^t) - g(W^t, x_t^*)$$

$$\nabla f(W, x) = (\nabla_W f(W, x), \nabla_x f(W, x)), \nabla g(W, x) = (\nabla_W g(W, x), \nabla_x g(W, x)). \quad (6)$$

**Lemma 1** *Let $(W, x)$ be a solution to the $\mathcal{BP}_\epsilon$. This point is also an $\epsilon$-accurate Pareto stationarity point for $\{l_i(x)\}$ satisfying*

$$\min_{w \in \mathcal{W}} \|G(x)w\|^2 = \mathcal{O}(\epsilon).$$

**Proof 1** *According to the definition of $\mathcal{BP}_\epsilon$, its solution $(W, x)$ satisfies that*

$$g(W, x) - g(W, x^*) \leq \epsilon. \tag{7}$$

*Further, according to Assumption 1, we can obtain*

$$g(W, x) \geq g(W, x^*) + \nabla_x g(W, x^*)(x - x^*) + \frac{1}{2L_g}\|\nabla_x g(W, x) - \nabla_x g(W, x^*)\|^2.$$

*Since $x^* \in \arg\min_x g(W, x)$ and $g(W, x) = \sum_{i=1}^{K} \sigma_i(W)l_i(x)$, we have $\nabla_x g(W, x^*) = 0$ and $\nabla_x g(W, x) = \sum_{i=1}^{K} \sigma_i(W)\nabla_x l_i(x) = G(x)\sigma(W)$ . We can obtain,*

$$\|G(x)\sigma(W)\|^2 \leq 2L_g(g(W, x) - g(W, x^*)) = \mathcal{O}(\epsilon), \tag{8}$$

*where the last inequality follows from eq. (7). Furthermore, since we have used softmax at the last layer of our neural network, $\sigma(W)$ belongs to the probability simplex $\mathcal{W}$. Thus, we can derive*

$$\min_{w \in \mathcal{W}} \|G(x)w\|^2 \leq \|G(x)\sigma(W)\|^2 = \mathcal{O}(\epsilon).$$

*Thus, the solution $(W, x)$ to the $\mathcal{BP}_\epsilon$ also satisfies Pareto stationarity of the loss functions $\{l_i\}$*

**Theorem 2 (Restatement of Theorem 1)** *Suppose Assumptions 1-2 are satisfied. Select hyperparameters*

$$\alpha \in (0, \frac{1}{L_f + \lambda(2L_g + L_g^2\mu)}], \ \beta \in (0, \frac{1}{L_g}], \lambda = L\sqrt{3\mu\epsilon^{-1}}, \ \text{and} \ N = \Omega(\log(\alpha t)).$$

*(i) Our method with the updates eq. (3) and eq. (4) (i.e., Algorithm 2 in the appendix) finds an $\epsilon$-accurate stationary point of the problem $\mathcal{BP}_\lambda$. If this stationary point is a local/global solution to $\mathcal{BP}_\lambda$, it is also a local/global solution to $\mathcal{BP}_\epsilon$. Furthermore, it is also an $\epsilon$-accurate Pareto stationary point for loss functions $l_i(x), i = 1, ..., K$.*

*(ii) Moreover, if $\|\nabla_W g(W^t, z_N^t)\| = \mathcal{O}(\epsilon)$ for $t = 1, ..., T$. The simplified method in Algorithm 1 also achieves the same convergence guarantee as that in $(i)$.*

**Proof 2** *We start with the first half of our theorem. Directly from Theorem 3 in Shen & Chen (2023), Algorithm 2 achieves an $\epsilon$-accurate stationary point of $\mathcal{BP}_\lambda$ with $\widetilde{\mathcal{O}}(\epsilon^{-1.5})$ iterations such that*

$$\frac{1}{T}\sum_{t=0}^{T-1}\|\nabla f(W^t, x^t) + \lambda(\nabla g(W^t, x^t) - \nabla g(W^t, x_t^*))\|^2 \leq \frac{F(W^0, x^0)}{\alpha T} + \frac{10L^2L_g^2}{T} = \mathcal{O}(\epsilon).$$

*Recall that $F(W^0, x^0) = f(W^0, x^0) + \lambda(g(W^0, x^0) - g(W^0, x_0^*))$. According to the Proposition 2 in Shen & Chen (2023) by setting $\delta = \epsilon$ therein, we can have $g(W^T, x^T) - g(W^T, x_T^*) \leq \epsilon$ if this stationary point is local/global solution to $\mathcal{BP}_\lambda$. Then by using Lemma 1, we know that this $\epsilon$-accurate stationary point is also an $\epsilon$-accurate Pareto stationary point of loss functions $\{l_i(x)\}$ satisfying*

$$\min_{w \in \mathcal{W}} \|G(x^T)w\|^2 = \mathcal{O}(\epsilon).$$

*The proof of the first half of our theorem is complete.*

*Then, for the second half, since we have built the connection between the stationarity of $\mathcal{BP}_\lambda$ and Pareto stationarity, we prove that the single-loop Algorithm 1 achieves an $\epsilon$-accurate stationary point of $\mathcal{BP}_\lambda$. Recall that*

$$\|\nabla f(W^t, x^t) + \lambda(\nabla g(W^t, x^t) - \nabla g(W^t, x_t^*))\|^2$$

$$\overset{(i)}{\leq} 2\|\nabla f(W^t, x^t) + \lambda\nabla g(W^t, x^t)\|^2 + 2\lambda^2\|\nabla g(W^t, x_t^*)\|^2$$

$$\overset{(ii)}{=} 2\|\nabla f(W^t, x^t) + \lambda \nabla g(W^t, x^t)\|^2 + 2\lambda^2 \|\nabla_W g(W^t, x_t^*)\|^2$$

$$\overset{(iii)}{\leq} 2\|\nabla f(W^t, x^t) + \lambda \nabla g(W^t, x^t)\|^2 + 4\lambda^2 \|\nabla_W g(W^t, x_t^*) - \nabla_W g(W^t, z_N^t)\|^2$$

$$+ 4\lambda^2 \|\nabla_W g(W^t, z_N^t)\|^2, \tag{9}$$

*where $(i)$ and $(iii)$ both follow from Young's inequality, and $(ii)$ follows from $\nabla_x g(W^t, x_t^*) = 0$. Besides, recall that $z_N^t$ is the intermediate output of the subloop in Algorithm 2. We next provide the upper bounds of the above three terms on the right-hand side (RHS). For the first term, we utilize the smoothness of $\Phi(\theta^t) = \nabla f(\theta^t) + \lambda \nabla g(\theta^t)$ where $L_\Phi = L_f + \lambda L_g$ and $\theta^t = (W^t, x^t)$.*

$$\Phi(\theta^{t+1}) \leq \Phi(\theta^t) + \langle \nabla \Phi(\theta^t), \theta^{t+1} - \theta^t \rangle + \frac{L_\Phi}{2}\|\theta^{t+1} - \theta^t\|^2$$

$$\overset{(i)}{\leq} \Phi(\theta^t) - \frac{\alpha}{2}\|\nabla \Phi(\theta^t)\|^2,$$

*where $(i)$ follows from $\alpha \leq \frac{1}{L_\Phi} = \mathcal{O}(\lambda^{-1})$. Thus, we can obtain*

$$\|\nabla \Phi(\theta^t)\|^2 \leq \frac{2}{\alpha}(\Phi(\theta^t) - \Phi(\theta^{t+1})). \tag{10}$$

*Then for the second term on the RHS in eq. (9), we follow the same step in the proof of Theorem 3 in Shen & Chen (2023) and obtain*

$$4\lambda^2 \|\nabla_W g(W^t, x_t^*) - \nabla_W g(W^t, z_N^t)\|^2$$

$$\leq 4\lambda^2 L_g^2 \mu \left(1 - \frac{\beta}{2\mu}\right)^N (g(W^t, x^t) - g(W^t, x_t^*))$$

$$\overset{(i)}{\leq} 4\lambda^2 L_g^2 \left(1 - \frac{\beta}{2\mu}\right)^N \|\nabla_x g(W^t, x^t)\|^2$$

$$= 4\lambda^2 L_g^2 \left(1 - \frac{\beta}{2\mu}\right)^N \left\|\frac{x^{t+1} - x^t + \alpha \nabla_x f(W^t, x^t)}{\alpha\lambda}\right\|^2$$

$$\overset{(ii)}{\leq} 8\lambda^2 L_g^2 \left(1 - \frac{\beta}{2\mu}\right)^N \left(\frac{\|\theta^{t+1} - \theta^t\|^2}{\alpha^2\lambda^2} + \frac{L^2}{\lambda^2}\right)$$

$$\overset{(iii)}{\leq} \frac{1}{2\alpha^2}\|\theta^{t+1} - \theta^t\|^2 + \frac{2L^2 L_g^2}{\alpha^2 t^2}$$

$$= \frac{1}{2}\|\nabla \Phi(\theta^t)\|^2 + \frac{2L^2 L_g^2}{\alpha^2 t^2}, \tag{11}$$

*where $(i)$ follows from the PL condition, $(ii)$ follows from Young's inequality and Assumption 1, and $(iii)$ follows from the selection on $N \geq \max\{-\log_{c_\beta}(16L_g^2), -2\log_{c_\beta}(2\alpha t)\}$ with $c_\beta = 1 - \frac{\beta}{2\mu}$. Lastly, for the last term at the RHS in eq. (9), we have,*

$$4\lambda^2 \|\nabla_W g(W^t, z_N^t)\|^2 = \mathcal{O}(\lambda^2 \epsilon^2), \tag{12}$$

*where this inequality follows from our experimental observation. Furthermore, substituting eq. (10), and eq. (11) into eq. (9) yields*

$$\|\nabla f(W^t, x^t) + \lambda(\nabla g(W^t, x^t) - \nabla g(W^t, x_t^*))\|^2$$

$$\leq \frac{5}{2}\|\nabla \Phi(\theta^t)\|^2 + \frac{2L^2 L_g^2}{\alpha^2 t^2} + 4\lambda^2 \|\nabla_W g(W^t, z_N^t)\|^2$$

$$\leq \frac{5}{\alpha}(\Phi(\theta^t) - \Phi(\theta^{t+1})) + \frac{2L^2 L_g^2}{\alpha^2 t^2} + 4\lambda^2 \|\nabla_W g(W^t, z_N^t)\|^2. \tag{13}$$

*Therefore, telescoping the above inequality yields,*

$$\frac{1}{T}\sum_{t=0}^{T-1} \|\nabla f(W^t, x^t) + \lambda(\nabla g(W^t, x^t) - \nabla g(W^t, x_t^*))\|^2$$

$$= \mathcal{O}\left(\frac{\lambda}{\alpha T} + \frac{1}{\alpha^2 T} + \lambda^2 \epsilon^2\right).$$

*According to the parameter selection that $\lambda = \mathcal{O}(\epsilon^{-\frac{1}{2}})$, $\alpha = \mathcal{O}(\epsilon^{\frac{1}{2}})$, and $T = \mathcal{O}(\epsilon^{-2})$, we can obtain*

$$\frac{1}{T}\sum_{t=0}^{T-1}\|\nabla f(W^t, x^t) + \lambda(\nabla g(W^t, x^t) - \nabla g(W^t, x_t^*))\|^2 = \mathcal{O}(\epsilon).$$

*Therefore, Algorithm 1 can achieve a stationary point of $\mathcal{BP}_\lambda$ with $\mathcal{O}(\epsilon^{-2})$ iterations. If this stationary point is a local/global solution to $\mathcal{BP}_\lambda$, it is also a solution to $\mathcal{BP}_\epsilon$ according to Proposition 2 in Shen & Chen (2023). Then, by using Lemma 1, we know this stationary point is also an $\epsilon$-accurate Pareto stationary point of the original loss functions. The proof is complete.*

## D  THE USE OF LARGE LANGUAGE MODELS (LLMS)

In the preparation of this manuscript, large language models (LLMs) were used only as writing aids to assist with language polishing and stylistic refinement. The extent of such use was limited, and all technical content, formulations, experimental designs, and conceptual contributions were developed by the authors. Importantly, LLMs were not used for ideation and methodology development.

