# OpenReview forum: "LDC-MTL: Balancing Multi-Task Learning through Scalable Loss Discrepancy Control"
_ICLR.cc/2026/Conference — Submitted to ICLR 2026_

### Official Review · Reviewer_PdRf · 2025-10-27

**Soundness:** 3
**Presentation:** 3
**Contribution:** 3
**Rating:** 6
**Confidence:** 4

**Summary:**

This paper investigates MTL from the perspective of loss balancing. Specifically, it proposes a bi-level loss balancing paradigm to dynamically adjust the task weights and promote the balanced optimization. Besides, it also proposes a first-order alternative for efficiency purposes. Extensive experiments have demonstrated the proposed method's ability to achieve competitive performance across multiple mainstream MTL datasets.

**Strengths:**

1. Employing a bi-level paradigm for MTL becomes popular recently, thus this paper is timely.
2. Developing efficient MTL algorithms is crucial in the era of large foundation models, and this paper addresses a timely and important issue.
3. Extensive evaluation demonstrates excellent performance across mainstream MTL datasets.

**Weaknesses:**

1. It remains unclear how Eqn. (1) facilitates balanced task learning speeds, especially given that $\tau$ is fixed at 1 in most experimental settings.
2. The paper lacks sufficient details regarding the router model, which plays a critical role in the proposed framework.
3. The proposed method adopts a different training schedule (e.g., learning rate, batch size) compared to prior MTL approaches [1][2]; please provide a discussion on GPU memory usage under this setting.
4. It is recommended to report another widely adopted MTL metric, Mean Rank (MR), for a more comprehensive evaluation [1][2].
5. Please specify which version of LDC-MTL was used for evaluation.

Reference:

[1] Fair Resource Allocation in Multi-Task Learning. ICML 2024.

[2] FAMO: fast adaptive multitask optimization. NeurIPS 2023.

**Questions:**

Please refer to the Weaknesses.

---

> ### Author Response · Authors · 2025-11-20
>
> **Weakness 1**
>
> **Answer:** Thank you for the question. To clarify, our objective is to reduce loss discrepancy rather than to directly control task learning speeds. Regulating the task training speeds would require access to all individual gradients $G=[g_1,g_2,...,g_K]$, and this leads to a high $\mathcal{O}(K)$ cost. Therefore, we do not take learning-speed control as our optimization target.
>
> If the reviewer is interested in this direction, a naive approach would be to reduce the discrepancy among the terms $g_i^\top Gw$ for all tasks, where $w$ is a weight vector and $i$ is the task index. We would leave it for future work.
>
> **Weakness 2**
>
> **Answer:** Thank you for pointing this out. In general, $\sigma(W)$ refers to a lightweight routing network parameterized by $W$, which takes as input the output features from the shared backbone of the model. While the router design varies slightly across the four datasets, it remains consistently simple and small in scale relative to the full model. Below, we provide the full details for all four datasets:
>
> Cityscapes: A 3-layer MLP maps 128-dimensional input features to 2 output weights, with a total of 10,402 parameters. The full SegNet model contains 41,217,386 parameters. NYU-v2: A 3-layer MLP maps 192-dimensional input features to 3 output weights, comprising 23,184 parameters. The SegNet model used here contains 44,229,652 parameters. QM9: A directly trainable weight vector of shape [11, 1] is used, corresponding to 11 tasks. The model contains 617,686 parameters in total. CelebA: A directly trainable weight vector of shape [40, 1] is used for the 40 classification tasks. The full model has 5,217,720 parameters.
>
> **Weakness 3**
>
> **Answer:** The training setup follows the same settings in [1][2] as mentioned in Section A.1 in the appendix. The GPU memory usage comparison is shown in the following table. Lastly, if we miss anything in your question, feel free to ask.
>
> **Table: GPU memory usage (GB) and running time (s) per epoch on the Cityscapes dataset**
>
> | Method    | Type        | GPU memory (GB) | Time/epoch (s) |
> |-----------|-------------|------------------|----------------|
> | FAMO      | O(1)        | 9.26             | 83             |
> | FairGrad  | O(K)        | 10.31            | 145            |
> | LDC-MTL   | O(1)        | 9.26             | 76             |
>
> **Weakness 4**
>
> **Answer:** Thanks for the suggestion. We have included the mean rank in Tables 6-7 in the appendix of the updated manuscript.
>
> **Weakness 5**
>
> **Answer:** All reported experimental results are obtained using the efficient variant of our method, corresponding to Algorithm 1 in the paper.

---

> > ### Comment · Reviewer_PdRf · 2025-11-26
> >
> > Thank you for the authors’ additional results and details. Some of my concerns have been addressed; however, I still have the following remaining issues:
> >
> > - Regarding the third weakness, my concern is that the training strategy used in this paper differs from that used in FAMO on both Cityscapes and NYUv2. For example:
> >
> >   - On Cityscapes, this paper uses lr = 3e-4, whereas FAMO uses lr = 1e-4.
> >
> >   - On NYUv2, this paper uses batch size = 8, whereas FAMO uses batch size = 2.
> >
> >    Such differences may influence performance comparison. I encourage the authors to clarify these choices or provide justification.
> >
> > - The authors have reported MR on NYUv2 and QM9, but please also report MR on all evaluated datasets (i.e., Cityscapes and CelebA). Furthermore, the reported results are less competitive compared to existing advanced MTL approaches. Some discussion on this might be needed.

---

> ### Author Response · Authors · 2025-11-26
>
> We sincerely thank the reviewer for the professional comments! Here are our clarifications. Hopefully they could address your concerns.
>
> **Question 1**
>
> **Answer:** Thanks for the careful check. For Cityscapes, we slightly tuned the learning rate to obtain optimal results for our method, while keeping all other training settings identical to the standard configuration. For NYUv2, the actual batch size used for NYUv2 is 2, consistent with FAMO. The value “8” in line 741 is a typo, and the correct setting is documented in the supplementary material.
>
> In summary, these minor differences do not influence the performance comparison, and we have revised the typo for clarity in the manuscript.
>
> **Question 2**
>
> **Answer:** We have added the mean rank for the CelebA and Cityscapes datasets in Tables 1-2 in the updated manuscript for completeness.
>
> We acknowledge that our method does not always achieve the absolute best performance, particularly when compared to the latest $\mathcal{O}(K)$ approaches such as ConsMTL. However, within the class of efficient and fast $\mathcal{O}(1)$ methods, our approach consistently delivers strong empirical results. When compared with many established $\mathcal{O}(K)$ methods, such as NashMTL, FairGrad, our method often surpasses their performance. It is also important to emphasize that computational cost is a critical factor in multi task learning, especially in large scale or resource constrained environments. Our approach offers the strongest performance among $\mathcal{O}(1)$ methods while maintaining constant computational overhead. We hope this provides a valuable contribution, especially in light of the rapidly increasing scale of modern models.

---

> > ### Comment · Reviewer_PdRf · 2025-11-27
> >
> > Thank you for your additional clarification. Such an efficient MTL algorithm might be applicable in today's foundation model era. However, the scalability of $\sigma(W)$ is still doubtful. Anyway, the authors have addressed my concerns; I would like to maintain my positive rating. Good luck.

---

> ### Author Response · Authors · 2025-11-27
>
> Dear reviewer PdRf,
>
> Thank you for the constructive feedback and for maintaining your positive rating. We appreciate your insight regarding the scalability of $\sigma(W)$. We will leave the exploration of applying our method and examining the scalability of our method in the foundation models for future work.
>
> Best,
>
> Authors

---

### Official Review · Reviewer_7Dov · 2025-10-30

**Soundness:** 2
**Presentation:** 2
**Contribution:** 2
**Rating:** 4
**Confidence:** 4

**Summary:**

This paper introduces LDC-MTL, a scalable multi-task learning method that tackles the persistent issue of loss discrepancy across task objectives. Motivated by the difficulty of balancing tasks whose losses operate on different scales or progress at different speeds, LDC-MTL formulates the learning challenge as a bilevel optimization problem: the lower-level optimizes a weighted sum of task losses, and the upper-level adjusts the weights to minimize their discrepancies. The authors propose a single-loop, first-order algorithm that claims $\mathcal{O}(1)$ memory and time overhead per iteration, compared to $\mathcal{O}(K)$ for previous gradient manipulation techniques. Experiments across standard MTL benchmarks demonstrate the efficiency and competitive accuracy of LDC-MTL.

**Strengths:**

1. The bilevel formulation provides a clear and principled avenue for directly controlling loss discrepancies while aiming for balanced performance across tasks (Section 4).

2. The authors conducted extensive experiments to verify the effectiveness of the proposed LDC-MTL.

3. The authors provide a thorough theoretical analysis, showing that the algorithm achieves $\epsilon$-Pareto stationarity under standard Lipschitz and PL conditions.

**Weaknesses:**

1. The authors claim that LDC-MTL achieves $\mathcal{O}(1)$ memory and time overhead per iteration. However, this property has already been stated in prior work such as FAMO. In fact, all loss-based MTL methods (e.g., UW, DWA, and more recently GO4Align) inherently possess this characteristic, as they aggregate task losses directly rather than computing gradients for each task separately—thus requiring only a single backward pass per iteration. Therefore, this advantage cannot be regarded as a distinctive contribution of the proposed method.

2. The idea of employing bilevel optimization for multi-task learning is not novel. Recent works, including GO4Align and ConsMTL [2], have already adopted similar formulations, making the conceptual contribution of this paper relatively incremental.

3. In Figure 1, the authors present the performance of LDC-MTL and other methods on a toy example. However, since there is no dominating solution along the Pareto front, convergence to the same Pareto stationary point cannot be considered a meaningful advantage in the absence of any prior preference among tasks.

4. The proposed method involves several hyperparameters, such as the learning rate $\alpha$ and the penalty coefficient $\lambda$. The authors determine these through grid search, which contradicts the claimed advantage of computational efficiency, as it implies multiple runs are needed to find suitable values. More thorough ablation studies and analyses are necessary to demonstrate the robustness of the method and to provide practical guidance for hyperparameter selection.

5. Confusing supplementary material. The authors appear to provide code in the supplementary material, yet upon inspection, there seems to be no actual implementation corresponding to the proposed LDC-MTL method. This raises concerns about reproducibility and transparency; the authors should ensure that the provided materials are complete and directly related to the presented work.

6. Missing references. The paper overlooks several key related works [1,2,3], some of which report superior empirical performance (though this does not affect my evaluation of the paper). In the MTL field, there are no universally accepted quantitative metrics for superiority—measures such as $\Delta m\\%$ and MR are only proxy indicators, and it is well known that $\Delta m\\%$ tends to favor baselines with lower initial performance. Therefore, I do not insist that the authors include these methods in their experiments, but they should at least acknowledge and differentiate them properly in the Related Work section.

[1] Independent component alignment for multi-task learning. CVPR 2023

[2] Towards Consistent Multi-Task Learning: Unlocking the Potential of Task-Specific Parameters. CVPR 2025

[5] Rep-MTL: Unleashing the Power of Representation-level Task Saliency for Multi-Task Learning. ICCV 2025

**Questions:**

Refer to weaknesses.

---

> ### Author Response · Authors · 2025-11-20
>
> **Weakness 1**
>
> **Answer:** We agree that achieving $\mathcal{O}(1)$ memory and time overhead per iteration is a property shared by loss-based MTL methods, and we do not regard this as a unique contribution of our work. Instead, our contribution is to retain the efficiency of $\mathcal{O}(1)$ methods while achieving performance comparable to or even better than these strong but more expensive gradient manipulation approaches (such as FairGrad, ConsMTL), as shown in our experiments.
>
> [a] On Penalty-based Bilevel Gradient Descent Method
> [b] Multi-objective meta-learning
> [c] A first-order multi-gradient algorithm for multi-objective bi-level optimization
>
> **Weakness 2**
>
> **Answer:** We admit that both GO4Align and ConsMTL have a bilevel optimization formulation, but there are fundamental differences in formulation, purpose, and implementation.
>
> Differences with GO4Align: In GO4Align, the lower-level problem is designed to identify \textbf{task groupings}, while the upper-level objective minimizes the weighted task loss. It updates the group assignment matrix implicitly within the lower-level optimization before updating model parameters. Lastly, our method provides a convergence analysis achieving a Pareto stationary point, whereas GO4Align does not offer such theoretical guarantees.
>
> Differences with ConsMTL: ConsMTL addresses gradient conflict by manipulating per-task gradients, with the upper level aggregating gradients and the lower level updating task-specific parameters. This leads to an $\mathcal{O}(K)$ overhead and lacks convergence guarantees to Pareto-stationary points. Our method, by contrast, maintains $\mathcal{O}(1)$ complexity and provides such guarantees.
>
> We have included this discussion in Section B in the appendix of the updated manuscript.
>
> **Weakness 3**
>
> **Answer:** The toy example in Figure 1 illustrates an imbalanced scenario where the two tasks exhibit different loss scales following FAMO. If the tasks have equal preference or importance, the most balanced solution should correspond to a point where both tasks receive similar updates (the middle point on the Pareto front). However, methods such as LS and MGDA converge to imbalanced solutions, and FAMO partially mitigates this effect. In contrast, our method explicitly corrects for the loss discrepancy and reaches the most balanced point.
>
> **Weakness 4**
>
> **Answer:** In our experiments, the penalty constant $\lambda$ does require some tuning effort, but the results remain relatively robust across a broad range of values. As shown in Table 4 and Table 9 in the appendix, the performance remains strong for a wide spectrum of $\lambda$, indicating that the method is not overly sensitive to this hyperparameter. In practice, a coarse search over [0.01,1] is typically sufficient to identify a good value. Regarding the step size $\alpha$, as noted in line 412, we simply follow the settings used in prior work and did not perform extensive tuning.
>
> **Weakness 5**
>
> **Answer:** Thank you for the comment. The current implementation is written directly within the main function of each trainer.py file. We will refactor the code by integrating the LDC-MTL implementation into a dedicated LDC-MTL class within weight\_methods.py to make the structure cleaner and easier to follow.
>
> **Weakness 6**
>
> **Answer:** Thanks for pointing this out. We have included the comparison and discussion with these related papers, especially ConsMTL, in the appendix of the updated manuscript.

---

> > ### Comment · Reviewer_7Dov · 2025-11-26
> >
> > Thanks for the authors’ response, which has addressed some of my concerns. I still have some further questions. The authors also acknowledge that achieving O(1) memory and time overhead per iteration is a property shared by loss-based MTL methods, and thus is not a contribution unique to this work. Compared with GO4Align, LDC-MTL achieves competitive but not strictly superior performance.
> >
> > My question is: when efficiency and performance are roughly comparable, what is the core advantage of LDC-MTL over GO4Align and FAMO? If the bilevel formulation for loss discrepancy control is mainly a methodological innovation, does this formulation bring any distinctive benefits compared with GO4Align and FAMO? Are there phenomena observed in the experiments that LDC-MTL can reveal but GO4Align and FAMO cannot? I believe that a deeper analysis of why this method works in essence, and what advantages it has over other loss-based MTL methods with similar performance, would provide a more insightful understanding.

---

> > > ### Author Response · Authors · 2025-11-26
> > >
> > > Thank you for the follow-up question. The core advantage of LDC-MTL lies in its explicit loss-discrepancy control, which can be directly observed in our experiments. For example, we report the statistics of task-loss values for LDC-MTL, GO4Align, FAMO, and LS in the following table. LDC-MTL achieves the lowest mean (0.189) and lowest standard deviation (0.133) among all methods. In comparison, GO4Align focuses on task grouping, and FAMO does not explicitly control loss discrepancies. As a result, they produce higher loss statistics (GO4Align: mean=0.238, std=0.154; FAMO: mean=0.231, std=0.141), though better than LS. This empirically demonstrates the distinctive benefit of our formulation: it reduces loss discrepancies more effectively and leads to better overall balance.
> > >
> > > **Table: Statistics of loss values for LDC-MTL, GO4Align, FAMO, and LS on the CelebA dataset.
> > > Lower mean and std indicate better and more stable performance.**
> > >
> > > | Method     | Mean ↓ | Std ↓ | Min ↓  | Max ↓  |
> > > |------------|--------|--------|--------|--------|
> > > | **LDC-MTL** | **0.189** | **0.133** | 0.029 | **0.538** |
> > > | GO4Align   | 0.238  | 0.154  | **0.026** | 0.660 |
> > > | FAMO       | 0.231  | 0.141  | 0.027 | 0.621 |
> > > | LS         | 0.287  | 0.195  | 0.032 | 0.729 |

---

> > > > ### Author Response · Authors · 2025-11-27
> > > >
> > > > In addition to the loss discrepancy control analysis, we also summarize the performance improvements of our method over FAMO and GO4Align across the four datasets. As shown in the table, our method outperforms both baselines in most cases. In particular, on Cityscapes, LDC-MTL improves the $\Delta m\%$ from 8.11/8.13 to -0.57. Overall, the results demonstrate that the improvements over FAMO and GO4Align are consistent and clear on most datasets.
> > > >
> > > > **Table: Comparison of Δm% values across four datasets for LDC-MTL, GO4Align, and FAMO.**
> > > >
> > > > | Method    | Cityscapes | NYU-v2 | QM9   | CelebA |
> > > > |-----------|------------|--------|--------|--------|
> > > > | GO4Align  | 8.11    | **-6.08** | 52.7  | 0.88  |
> > > > | FAMO      | 8.13       | -4.10 | 58.5  | 1.21  |
> > > > | **LDC-MTL** | **-0.57** | -4.40 | **49.5** | **-1.31** |

---

> > > > > ### Comment · Reviewer_7Dov · 2025-11-27
> > > > >
> > > > > Thanks for the authors’ further response. Although I believe that both the experimental results and the presentation of the claimed contributions in the main text require further improvement, I agree that the idea of controlling loss discrepancy is of interest to the MTL community. Therefore, I will maintain a neutral stance toward this submission, but I will raise my score to 6 and leave the final decision to the Area Chair.

---

> > > > > > ### Author Response · Authors · 2025-11-27
> > > > > >
> > > > > > Thank you for reviewing our additional response and raising your score. We appreciate your feedback and will continue improving the clarity and presentation of our contributions in the main text based on all reviewers’ comments.

---

### Official Review · Reviewer_shRo · 2025-10-31

**Soundness:** 3
**Presentation:** 3
**Contribution:** 3
**Rating:** 4
**Confidence:** 3

**Summary:**

This work proposed a first-order bi-level optimization approach to solve the multi-task learning problem. The proposed method mainly contains two components: (i) a bilevel formulation for fine-grained loss discrepancy control, and (ii) a scalable first-order bilevel algorithm that requires only constant order time and memory.

**Strengths:**

1.  The proposed method avoids solving the complex bi-level structure in BLO. It also achieves good performance in MTL.
2. The paper structure is clear and easy to follow.
3. A theoretical proof to support the effectiveness of the proposed method.

**Weaknesses:**

1. Lacking $\mathcal{O}(1)$ baselines such as "Smooth Tchebycheff Scalarization for Multi-Objective Optimization, ICML 2024."

2. Using bi-level optimization to solve the MTL problem is not new. The "related work" part does not mention those works, such as "Multi-objective meta-learning, AIJ" and "A first-order multi-gradient algorithm for multi-objective bi-level optimization, ECAI 2024".

3. The proposed approach is quite like a penalty-based bi-level optimization approach, such as "On Penalty-based Bilevel Gradient Descent Method". Though the author cites some papers  (Kwon et al., 2023; Yang et al., 2023), but does not discuss the differences and how similar these approaches are. This is important for evaluating the theoretical contributions of this paper. In my opinion, the theoretical result does not show an advantage.

**Questions:**

See the weaknesses part.

---

> ### Author Response · Authors · 2025-11-20
>
> **Weakness 1**
>
> **Answer:** Thank you for pointing this out. A recent $\mathcal{O}(1)$ algorithm, STCH, is compared with our method on the QM9 dataset in the following Table. It has comparable performance with FairGrad and even GO4Align. We have also cited this paper and added a comparison.
>
> **Table: Detailed results on QM9 (11-task) dataset of STCH and LDC-MTL**
>
> | Method    | μ | α | ε_HOMO | ε_LUMO | R² | ZPVE | U₀ | U | H | G | c_v | Δm% |
> |-----------|----|------|---------|---------|------|------|------|------|------|------|------|-------|
> | STL       | 0.067 | 0.181 | 60.57 | 53.91 | 0.502 | 4.53 | 58.8 | 64.2 | 63.8 | 66.2 | 0.072 |  |
> | GO4Align  | 0.17 | 0.35 | 102.4 | 119.0 | 1.22 | 4.94 | 53.9 | 54.3 | 54.3 | 53.9 | 0.11 | 52.7 |
> | STCH      | 0.166 | 0.260 | 94.48 | 101.2 | 1.850 | 4.88 | 58.34 | 58.68 | 58.70 | 58.27 | 0.104 | 56.9 |
> | FairGrad  | 0.117 | 0.253 | 87.57 | 84.00 | 2.15 | 5.07 | 70.89 | 71.17 | 71.21 | 70.88 | 0.095 | 57.9 |
> | LDC-MTL   | 0.23 | 0.29 | 123.89 | 111.95 | 0.97 | 3.99 | 42.73 | 43.1 | 43.2 | 43.1 | 0.097 | 49.5 |
>
> **Weakness 2**
>
> **Answer:** Thank you for pointing out these relevant works. We would like to clarify how our approach differs. In short, our method differs from [1–2] in formulation, purpose, and computational complexity.
>
> First, the formulations are fundamentally different. In [1–2], each task solves a hierarchical objective in a meta-learning setting, whereas each task in our setting solves a single-level objective function in a multi-task learning setting. Second, the goals differ. The methods in [1–2] aim to learn better meta-learners, whereas our work focuses on addressing the imbalanced learning problem in MTL through loss-discrepancy control. Lastly, both MOML and FORUM rely on double-loop optimization with
> $\mathcal{O}(K)$ computational overhead. In contrast, our proposed LDC-MTL introduces a fully single-loop bilevel algorithm that achieves $\mathcal{O}(1)$ per-iteration time and memory complexity.
>
> We have included these papers in the related works.
>
> [1] Multi-objective meta-learning
> [2] A first-order multi-gradient algorithm for multi-objective bi-level optimization
>
> **Weakness 3**
>
> **Answer:** Thank you for your comment. While we agree that our method adopts a penalty-based bi-level optimization approach, our Algorithm 1 has adjustments based on experimental exploration.
>
> First, as discussed in Section 4.3, both [3–4] require a sub-loop to approximate the lower-level optimum $x_t^*$. In contrast, our empirical analysis in Figure 3 shows that this approximation can be safely eliminated, allowing Algorithm 1 to operate as a fully single-loop method.
>
> In terms of the analysis, the lower-level problem in [3] has to be assumed to be strongly convex, while we assume a relaxed PL condition. Meanwhile, although [4] considers the PL condition, its analysis is under a single-task setting, and there is no analysis of whether the penalty-based method converges to Pareto stationary points in MOO problems. Therefore, we establish a connection between the stationarity of $BP_\lambda$ and Pareto stationarity. Moreover, with a gradient-vanishing assumption, the adjusted Algorithm can also attain an $\epsilon$-accurate stationary point for the problem $BP_\lambda$ and an $\epsilon$-accurate Pareto stationary point for the original loss functions.
>
> This discussion has been included in Section C.1 in the appendix of the updated manuscript and will be merged into the main text.
>
> [3] A Fully First-Order Method for Stochastic Bilevel Optimization
> [4] On Penalty-based Bilevel Gradient Descent Method

---

### Official Review · Reviewer_xsvW · 2025-11-01

**Soundness:** 3
**Presentation:** 3
**Contribution:** 3
**Rating:** 6
**Confidence:** 4

**Summary:**

The paper proposes LDC-MTL, a scalable loss discrepancy control method for multi-task learning (MTL). It formulates MTL as a bilevel optimization problem, where the lower level minimizes a weighted sum of task losses and the upper level minimizes the discrepancies among those losses. A first-order, single-loop approximation is introduced, leading to an $\mathcal{O}(1)$ time and memory algorithm that scales with the number of tasks. Theoretical analysis establishes convergence both to a stationary point of the bilevel problem and to an $\epsilon$-accurate Pareto-stationary point. Experiments on CelebA ($40$ tasks), QM9 ($11$ tasks), NYU-v$2$ ($3$ tasks), and Cityscapes ($2$ tasks) show competitive or superior $\Delta m\\%$ compared to scalarization and gradient-manipulation baselines, with runtime close to linear scalarization (LS).

**Strengths:**

The paper's main strengths lie in its clear bilevel formulation for controlling task-wise loss discrepancies, which provides a principled alternative to existing loss balancing methods. The proposed approach is both scalable and lightweight, featuring a single-loop update that avoids the typical $\mathcal{O}(K)$ gradient storage overhead found in many multi-task algorithms. Theoretical analysis further reinforces the work by providing convergence guarantees that ensure Pareto stationarity under mild assumptions. Empirically, the paper presents comprehensive experiments across four benchmark datasets and compares against competitive baselines, demonstrating consistent improvements. Moreover, the additional analyses on gradient conflict reduction and loss variance offer valuable insights into the behavior and interpretability of the proposed method.

**Weaknesses:**

1. The claimed $O(1)$ efficiency critically depends on the empirical observation that $||\nabla_W g(W^t,z^t_N)||$ remains small. While this is illustrated for the Cityscapes dataset $(K = 2)$, providing similar empirical evidence for other datasets, particularly CelebA $(K = 40)$, would strengthen the generality of this claim.

2. The experimental analyses on loss discrepancy and gradient conflict (Table $3$, Figures $6$ and $7$) are conducted only against linear scalarization. For a fairer assessment, comparisons should also include the strongest loss balancing baselines, such as GO4Align, which achieves top performance in the main experiments.

**Questions:**

1. Could you provide a performance comparison between Algorithm $1$ and Algorithm $2$? Additionally, what do you think of a hybrid strategy that runs Algorithm $2$ until $||\nabla_W g(W^t, z^t_N)||$ drops below a threshold relative to $||\nabla_W g(W^t, x^t)||$, and then switches to Algorithm $1$ for the remainder of training?

2. Could you include a comparison with ConsMTL $[1]$, a recent state-of-the-art gradient-manipulation method that formulates MTL as a bi-level optimization over shared and task-specific parameters? This would strengthen the claim that LDC-MTL consistently outperforms gradient-based baselines.

3. In Appendix A-$7$, it is stated that the scatter plots in Figures $5 (a)$ and $5 (b)$ follow the same setting as Figure $8$ in $[2]$. Does this imply that the axis labels in Figures $5 (a–b)$ might have been mislabeled? Additionally, could you include FAMO and GO4Align (each trained with distinct random seeds) in this Pareto front analysis for a more comprehensive comparison?

$[1]$ Qin, Xiaohan, Xiaoxing Wang, and Junchi Yan. "Towards Consistent Multi-Task Learning: Unlocking the Potential of Task-Specific Parameters." Proceedings of the Computer Vision and Pattern Recognition Conference. 2025.

$[2]$ Xin, Derrick, et al. "Do current multi-task optimization methods in deep learning even help?." Advances in neural information processing systems 35 (2022): 13597-13609.

---

> ### Author Response · Authors · 2025-11-20
>
> **Weakness 1**
>
> **Answer:** Thank you for the suggestion. We have added corresponding empirical evidence in Figure 7 in the appendix of the updated manuscript by visualizing the gradient-norm evolution on the CelebA dataset. Consistent with our observations on the Cityscapes dataset, the ratio $\\|\nabla_Wg(W^t,x^t)\\|/\\|\nabla_Wg(W^t,z_N^t)\\|$ remains substantially large throughout training. This confirms that the term $\\|\nabla_Wg(W^t,z_N^t)\\|$ is indeed negligible in practice, supporting the generality of our approximation.
>
> **Weakness 2**
>
> **Answer:** Thanks for the comment. We have added an analysis of both loss discrepancy and gradient conflict for GO4Align in Table 3, Figure 8, and Figure 9 in the appendix of the updated manuscript. First, for loss discrepancy analysis, GO4Align shows moderate mean and standard deviation values, higher than LDC-MTL but lower and more stable than LS. Besides, GO4Align attains the smallest minimum loss value. Regarding gradient conflict, GO4Align is able to reduce conflict to some extent, although the degree of mitigation is similarly moderate.
>
> **Question 1**
>
> **Answer:** Thank you for the thoughtful question. We agree that a hybrid strategy may be beneficial conceptually. However, it also raises several practical challenges. First, it introduces additional hyperparameters, including the inner-loop length $N$ and the switching threshold, which can vary significantly across datasets and architectures. Second, to obtain a sufficiently accurate approximation of the $x_t^*$, $N$ often needs to be large. If the threshold is difficult to satisfy, the resulting computational overhead may exceed $\mathcal{O}(K)$, defeating the purpose of the efficient design. Lastly, switching between optimization rules during training can introduce instability, especially in multi-task learning, where gradients do not interact trivially.
>
> Meanwhile, to further address the reviewer’s question, we conducted a preliminary experiment using a simple hybrid strategy: we apply Algorithm 2 with $N=3$ for the first 50 epochs and then switch to Algorithm 1. The results (shown in the table below) indicate that such a hybrid method has potential, but its performance is not SOTA. Further gains would likely require careful tuning and a more principled design of the switch criterion.
>
> **Table: Results on Cityscapes (2-task) dataset**
>
> | Method            | mIoU | Pix Acc | Abs Err | Rel Err | Δm% |
> |-------------------|------|---------|---------|---------|------|
> | STL               | 74.01 | 93.16 | 0.0125 | 27.77 |  |
> | GO4Align          | 72.63 | 93.03 | 0.0164 | **27.58** | 8.11 |
> | FairGrad          | **75.72** | **93.68** | 0.0134 | 32.25 | 5.18 |
> | LDC-MTL           | 74.53 | 93.42 | **0.0128** | **26.79** | **-0.57** |
> | LDC-MTL-hybrid    | **76.14** | **93.72** | **0.0129** | 32.76 | **4.47** |
>
> **Question 2**
>
> **Answer:** Thank you for pointing this out. ConsMTL is indeed a concurrent state-of-the-art $\mathcal{O}(K)$ gradient-manipulation approach that also adopts a bilevel formulation. Empirically, ConsMTL achieves the best performance on the QM9 dataset. Meanwhile, both methods obtain comparable performance on Cityscapes and CelebA datasets in the following tables, but ConsMTL consumes over 2-10$\times$ more time on these benchmarks.
>
> **Table: Results on Cityscapes (2-task) dataset of LDC-MTL and ConsMTL**
>
> | Method    | mIoU | Pix Acc | Abs Err | Rel Err | Δm% | Time/epoch (s) |
> |-----------|------|---------|---------|---------|------|-----------------|
> | STL       | 74.01 | 93.16 | 0.0125 | 27.77 |  |  |
> | ConsMTL   | 75.57 | 93.32 | 0.0131 | 26.41 | -0.59 | 171 |
> | LDC-MTL   | 74.53 | 93.42 | 0.0128 | 26.79 | -0.57 | 76 |
>
> **Table: Results on CelebA (40-task) dataset of LDC-MTL and ConsMTL**
>
> | Method    | Δm% | Time/epoch (s) |
> |-----------|------|----------------|
> | ConsMTL   | -1.42 | 987 |
> | LDC-MTL   | -1.31 | 96 |
>
> **Question 3**
>
> **Answer:** Thank you for pointing this out. The axis labels in Figures 5(a) and 5(b) were indeed incorrect, and we have corrected them in the revised version. In addition, we have included the results of FAMO and GO4Align, each trained with multiple random seeds, in Figures 5(a–b). These additional runs show that our method consistently provides dominating solutions over LS, FAMO, and GO4Align.

---

> > ### Comment · Reviewer_xsvW · 2025-11-25
> >
> > The authors have made a clear effort to address the concerns I raised in the initial review, including adding a comprehensive comparison with ConsMTL across all four datasets and responding thoroughly to the other points I mentioned. The updated results show that, according to the primary evaluation metric $\Delta m\\%$, ConsMTL outperforms LDC-MTL on Cityscapes, NYU-v2, QM9, and CelebA, with only marginal gaps on the 2-task and 40-task settings. At the same time, the authors report that ConsMTL incurs substantially higher computational cost, often $2\times$ to $10\times$ slower, thereby highlighting a meaningful efficiency–performance trade-off.
> > This strengthens the empirical section and clarifies the positioning of LDC-MTL as a practical, scalable $\mathcal{O}(1)$ alternative, even if it is not the top performer in $\Delta m\\%$. A minor issue that remains is that Table 1 in the main text still omits the ConsMTL results, which appear only in the appendix.
> >
> > Overall, the revision improves the paper’s clarity and completeness. *I do not raise the score* because the new experiments show that LDC-MTL does not achieve state-of-the-art $\Delta m\\%$ performance on any of the evaluated datasets, reducing the strength of its empirical superiority claims. *I also do not lower the score*, because the method is theoretically sound, computationally attractive, and now fairly positioned relative to strong baselines, making it a valuable and scalable contribution to multi-task learning. I therefore maintain my original recommendation and believe the paper would be a valuable addition to the conference.

---

> ### Author Response · Authors · 2025-11-25
>
> Dear Reviewer xsvW,
>
> Thank you very much for your thoughtful updates. We’re glad to hear that our responses have addressed your concerns. We also appreciate your constructive suggestions, which have greatly improved the quality and clarity of our paper. Thanks!
>
> Best regards,
>
> Authors

---

> > ### Author Response · Authors · 2025-11-25
> >
> > Dear Reviewer xsvW,
> >
> > Regarding Table 1, we have added the ConsMTL results back into the main text for completeness. In addition, Figure 4 reports the updated training time comparison, including ConsMTL. It can be seen that ur LDC-MTL remains competitive for its scalability.
> >
> > Thanks again!
> >
> > Best,
> > Authors

---

### Meta-Review · Area_Chair_hbAz · 2026-01-02

**Summary:**

This paper presents LDC-MTL, a novel approach for Multi-Task Learning (MTL). The four reviewers have raised significant, critical, and interconnected concerns that collectively challenge the paper's ‌novelty, empirical thoroughness, technical clarity, and scholarly rigor.‌

**Overarching Challenge to Novelty and Contribution:‌**\
This is the most critical and recurring theme. ‌**Reviewers 2 and 3 fundamentally question the paper's core claims.** Reviewer 3 argues that the **O(1) efficiency‌** is not a novel contribution but a standard property of loss-based methods. Both Reviewers 2 and 3 also challenge the novelty of **using ‌bi-level optimization‌ for MTL**, citing specific recent works (GO4Align, ConsMTL, and others) that have adopted similar formulations. The assertion that the proposed method closely resembles ‌penalty-based bi-level optimization‌ (Reviewer 2) without clear differentiation further undermines the perceived theoretical contribution.

**Insufficient and Potentially Unfair Empirical Validation:‌**\
The experimental section is found lacking by multiple reviewers. The analyses on loss discrepancy and gradient conflict are critiqued for only comparing against a weak baseline (**‌linear scalarization**‌), omitting stronger competitors like ‌GO4Align‌ (Reviewer 1). There is a consensus that **key ‌baselines are missing**‌ from comparisons, including ConsMTL (Reviewer 1), "Smooth Tchebycheff Scalarization" (Reviewer 2), and other cited SOTA methods (Reviewer 3). The ‌**toy example's advantage is deemed questionable**‌ (Reviewer 3). Reviewer 4 suggests **adding the ‌Mean Rank (MR)‌ metric** for a fairer evaluation. The empirical claims of superiority are therefore not fully substantiated.

**Technical Gaps and Reproducibility Concerns:**\
Several technical aspects require urgent attention. The ‌balancing mechanism‌ of Eqn. (1) is unclear (Reviewer 4). Vital details about the ‌router model‌ are missing (Reviewer 4). The use of ‌grid search‌ for hyperparameter tuning is seen as contradictory to the computational efficiency claim, necessitating robustness analysis (Reviewer 3). Most critically, ‌Reviewer 3 raises a severe red flag regarding reproducibility‌, stating that the supplementary code may not contain the actual LDC-MTL implementation. This is a fundamental issue that must be resolved.

**Deficiencies in Scholarly Presentation and Completeness:‌**\
The paper falls short in academic thoroughness. The ‌Related Work section is identified as incomplete‌ by both Reviewers 2 and 3, failing to acknowledge pertinent prior art in bi-level optimization and recent MTL literature. Potential ‌mislabeling in figures‌ needs checking (Reviewer 1). A discussion on ‌GPU memory‌ impact due to the altered training schedule is requested (Reviewer 4).

In conclusion, while the core idea of this paper may have merit, the concerns of four reviewers regarding novelty, experimental completeness, and reproducibility are substantial. A diligent and extensive revision addressing these points is essential for this work to make a convincing contribution to the MTL community.

**Reviewer Concerns:**

1. Concerns on Novelty and Contribution: Reviewer 3 argues that the **O(1) efficiency‌** is not a novel contribution. Both Reviewers 2 and 3 also challenge the novelty of **using ‌bi-level optimization‌ for MTL**. These concerns were not addressed after the rebuttal stage.

2. Concerns on Insufficient Evaluation: All four reviewers raised concerns on insufficient evaluation with missing baselines or metics. After the rebuttal stage, the proposed method is shown to be **inferior to GO4Align** on  NYU-v2 among O(1) methods in Table 1. In Figure 4, ConsMTL has only 11 times of running time on CelebA, which **does not increase linearly with K=40**. Considering that ConsMTL  significantly outperforms the proposed method,  it may be a better choice when both effency and effetiveness are concerned.

**Reviewer Scores:**

Revewers 1, 3 and 4 have participated in the discussion, and the final decison has been made based on their new comments. Reviewer 2 challenged the novelty of  using ‌bi-level optimization‌ for MTL, which was not addressed by the rebuttal.

---

### Decision · Program_Chairs · 2026-01-26

Reject